# A differentiable brain simulator bridging brain simulation and brain-inspired computing

**Chaoming Wang**[1*]**, Tianqiu Zhang**[1*]**, Sichao He**[2]**, Hongyaoxing Gu**[3]**, Shangyang Li**[1]**, Si Wu**[1†]

[1]School of Psychological and Cognitive Sciences, IDG/McGovern Institute for Brain Research,
Bejing Key Laboratory of Behavior and Mental Health, Peking-Tsinghua Center for Life Sciences,
Center of Quantitative Biology, Academy for Advanced Interdisciplinary Studies, Peking University
[2]School of Software Engineering, Beijing Jiaotong University
[3]Institute of Software, Chinese Academy of Sciences
[*]The authors contributed equally        [†] Corresponding author
{wangchaoming,syli,siwu}@pku.edu.cn,tianqiuakita@stu.pku.edu.cn

## Abstract

Brain simulation builds dynamical models to mimic the structure and functions of the brain, while brain-inspired computing (BIC) develops intelligent systems by learning from the structure and functions of the brain. The two fields are intertwined and should share a common programming framework to facilitate each other's development. However, none of the existing software in the fields can achieve this goal, because traditional brain simulators lack differentiability for training, while existing deep learning (DL) frameworks fail to capture the biophysical realism and complexity of brain dynamics. In this paper, we introduce BrainPy, a differentiable brain simulator developed using JAX and XLA, with the aim of bridging the gap between brain simulation and BIC. BrainPy expands upon the functionalities of JAX, a powerful AI framework, by introducing complete capabilities for flexible, efficient, and scalable brain simulation. It offers a range of sparse and event-driven operators for efficient and scalable brain simulation, an abstraction for managing the intricacies of synaptic computations, a modular and flexible interface for constructing multi-scale brain models, and an object-oriented just-in-time compilation approach to handle the memory-intensive nature of brain dynamics. We showcase the efficiency and scalability of BrainPy on benchmark tasks, and highlight its differentiable simulation for biologically plausible spiking models.

## 1 Introduction

Brain simulation aims to elucidate brain functions by building dynamical models that mimic the structure and dynamics of the brain (Gerstner et al., 2014), while brain-inspired computing aims to develop intelligent systems by learning from the structure and computational principles of the brain (Mehonic & Kenyon, 2021). The two fields are intertwined and their developments can facilitate each other. For example, brain simulation can provide BIC with models of neurons, synapses, networks, and inspirational information processing principles; while BIC can provide brain simulation with efficient algorithms for optimizing model parameters, simulation tools for running large-scale networks, and a testing bed for validating hypothesized neural mechanisms. Ideally, the two fields should share a common programming framework, so that they can benefit from each other's development by sharing models, mathematical tools, and emerging findings.

However, up to now, none of the existing software in the two fields can fully achieve this goal. Traditional brain simulators, such as NEURON (Hines & Carnevale, 1997), NEST (Gewaltig & Diesmann, 2007), and Brian2 (Goodman & Brette, 2008; Stimberg et al., 2019), are designed for simulating brain dynamics models with high fidelity and accuracy. They rely on customized numerical solvers and data structures that are not compatible with automatic differentiation, and hence cannot support training models with standard gradient-based methods. On the other hand, by leveraging the automatic differentiation functionality of deep learning (DL) frameworks like PyTorch (Paszke et al., 2019) and TensorFlow (Abadi et al., 2016), existing BIC libraries, such as snnTorch (Eshraghian et al.,

2021), Norse (Pehle & Pedersen, 2021), and SpikingJelly (Fang et al., 2020), provide convenient interfaces for building and training spike neural networks (SNNs). They are, however, not designed to capture the unique and important features of brain dynamics, and hence are not suitable to simulate large-scale brain networks with realistic biophysical properties.

In this paper, we propose BrainPy as an innovative solution to bridge this gap. Unlike traditional brain simulators, BrainPy leverages the power of the JAX (Frostig et al., 2018), allowing seamless integration with AI models. However, BrainPy goes beyond integration and introduces dedicated optimizations that unleash the full potential of a flexible, efficient, and scalable brain simulator within the JAX ecosystem. To capture the sparse and event-driven nature of brain computation, BrainPy provides a wide range of customized primitive operators. For enhanced flexibility in model construction across various brain organization scales, BrainPy offers a modular and composable interface. To handle the complexity of synaptic computations, BrainPy introduces a novel abstraction for executing diverse synaptic projections. Additionally, to tackle the memory-intensive demands of brain dynamics, BrainPy employs an object-oriented just-in-time (JIT) compilation approach. Leveraging the automatic differentiation capabilities of JAX, BrainPy represents a unique differentiable brain simulator that bridges the gap between brain simulation and BIC fields. We demonstrate the efficiency and scalability of BrainPy on several brain simulation and BIC tasks and showcase its ability to train biologically plausible spiking models with differentiability.

## 2 Related Works

**Brain Simulators.** Different brain simulators normally have different focuses. NEURON (Hines & Carnevale, 1997) allows users to define detailed biophysical models of neurons and synapses, with complex morphology and ion channels. NEST (Gewaltig & Diesmann, 2007) focuses on large-scale network models of point neurons and synapses, with simplified dynamics and connectivity patterns. Brian2 (Stimberg et al., 2014; 2019) targets being flexible and intuitive, allowing users to easily define dynamical models, environment interactions, and experimental protocols. Currently, the dominant programming approach in brain simulation is descriptive language (Blundell et al., 2018), by which users can use text (Stimberg et al., 2019; Vitay et al., 2015), JSON (Dai et al., 2020; Dura-Bernal et al., 2019), or XML (Gleeson et al., 2010) files to describe the model, and then translate the descriptive information into high-efficient C++ or CUDA code. The main advantage of this approach is the clear decoupling between mathematical description from its implementation details. However, this advantage comes with expensive costs, which include, for instance, the lack of flexibility and generality of defining new models not covered by the predefined constructs and functions of the custom language, the difficulty of integrating and interfacing with other tools and frameworks not using the same format, and the high learning cost for the unfamiliar syntax of the custom language. These limitations prevent the application of existing brain simulators to BIC models.

**BIC Libraries.** SNNs are the current dominating models in BIC for their advantages in biological interoperability and energy efficiency. A number of programming libraries have been developed for SNNs, such as NengoDL (Rasmussen, 2018), BindsNet (Hazan et al., 2018), snnTorch (Eshraghian et al., 2021), Norse (Pehle & Pedersen, 2021), SpikingJelly (Fang et al., 2020), and BrainCog (Zeng et al., 2022). These libraries utilize DL frameworks, such as PyTorch (Paszke et al., 2019) and TensorFlow (Abadi et al., 2016), to achieve the training of SNNs on various tasks that cannot be done in traditional brain simulators. So far, BIC libraries have mainly focused on the combination of spiking neurons with DL models, e.g., spiking convolutional neural networks and spiking recurrent neural networks. However, these libraries fall short of high-fidelity brain simulation. First, DL frameworks lack the dedicated components for sparse, event-driven, and scalable computation required for brain dynamics models. Second, BIC libraries designed for machine learning tasks often lack the essential capabilities to support realistic neuronal and synaptic simulations based on experimental data. The brain encompasses intricate biochemical and biophysical processes that span vast scales in both space and time. Unfortunately, current BIC libraries without these dedicated optimizations face significant challenges in accurately modeling such complex biophysical characteristics.

## 3 The Design of BrainPy

BrainPy is designed to take the combined advantages of *brain simulators* and *DL frameworks*. This innovative tool is specifically engineered to leverage the strengths of JAX (Frostig et al., 2018), a

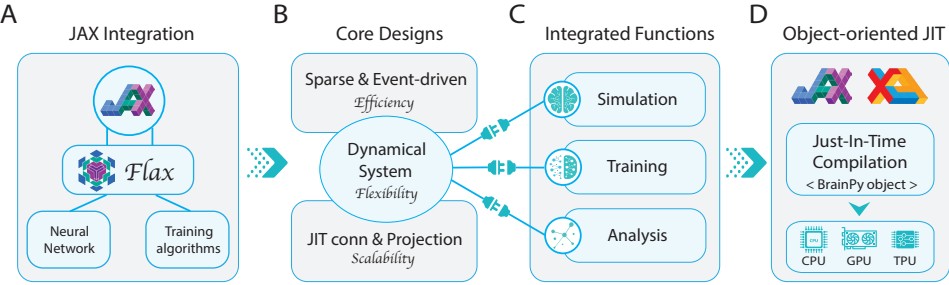

Figure 1: The overview of BrainPy architecture.

powerful AI framework, while simultaneously expanding upon it to enable flexible, efficient, and scalable simulation of various brain dynamics models.

By building upon JAX, BrainPy inherits the robust computational capabilities and extensive library support that JAX provides (see Figure 1A). This enables seamless integration of DL techniques, such as neural network architectures and gradient-based optimization algorithms, into its brain simulation framework. BrainPy facilitates smooth interoperation with existing JAX-based machine learning libraries (Babuschkin et al., 2020). Users can easily transform models from other libraries into BrainPy objects. For example, by utilizing the `brainpy.dnn.FromFlax` functionality, a Flax (Heek et al., 2020) model can be treated as a BrainPy module. Conversely, users can convert BrainPy models into formats compatible with other libraries. The `brainpy.dnn.ToFlax` feature, for instance, enables the interpretation of a dynamical system in BrainPy as a Flax recurrent cell, allowing brain models developed in BrainPy to be utilized within a machine-learning context.

However, BrainPy goes beyond being a mere extension of JAX. It introduces novel and fundamental features to empower users to simulate and analyze the intricate dynamics of the brain (refer to Figure 1B and Section 4). (1) BrainPy provides **flexibility** in modeling brain dynamics. The brain exhibits a unique multi-scale organization characterized by hierarchical modularity. To address this, BrainPy offers a modular and composable interface specifically designed for brain dynamics programming. With this interface, users can easily define and customize complex brain models across multiple levels of organization, from low-level ion channels, neurons, and synapses to high-level networks and systems. (2) BrainPy prioritizes **efficiency** in simulating brain dynamics. Brain dynamics involve unique properties such as event-driven computation and sparse connections. BrainPy abstracts these operations into primitive operators and gives a full set of primitive operators that achieve major speedups, from two to five orders of magnitude faster than traditional dense operators. (3) BrainPy tackles the **scalability** challenges of the brain's large-scale structure. The brain consists of massively interconnected neurons arranged in complex networks and circuits. To handle this, BrainPy offers specialized support, including just-in-time connectivity operators with zero memory footprint, automatic synapse merging for network topology optimization, and parallelization strategies for distributed computing.

Leveraging its distinctive brain simulation capabilities and seamless integration with DL techniques, BrainPy offers an integrated platform for comprehensive simulation, training, and analysis of brain dynamics models (see Figure 1C). Firstly, it enables efficient simulation of models at various scales of organization. These simulations can be executed in parallel across multiple threads, processors, and devices, facilitating parameter explorations and enhancing performance. Secondly, BrainPy supports the training of diverse model types based on neural data or behavioral tasks. For example, reservoir computing models can be trained using BrainPy's online and offline learning algorithms, and detailed spiking neural networks and rate-based recurrent neural networks can be trained using its back-propagation-based algorithms. Thirdly, BrainPy offers automatic analysis interfaces that unravel the underlying mechanisms of simulated or trained models. For instance, low-dimensional analyzers can perform phase plane and bifurcation analysis. On the other hand, high-dimensional analyzers facilitate slow point computation and linearization analysis for high-dimensional systems. Lastly, BrainPy achieves exceptional running speed through the utilization of JIT compilation. Through a novel object-oriented JIT transformation, BrainPy enables whole-graph optimization of class-based models into executable processes using JAX and XLA on CPU, GPU, or TPU devices during simulation, training, and analysis tasks (see Figure 1D).

## 4 METHODOLOGY

In this section, we present the specific optimizations implemented in BrainPy for brain modeling, and highlight the advancements achieved at the operator, model, and compilation levels to facilitate its goal of unifying models in brain simulation and BIC.

### 4.1 DEDICATED OPERATORS FOR SPARSE AND EVENT-DRIVEN COMPUTATION

Compared to DNN models, brain dynamics models perform computation in a different way. They usually have sparse connections (neurons in a network are typically interconnected with a probability smaller than 0.2 (Potjans & Diesmann, 2012)) and perform event-driven computations (synaptic states are updated only when the presynaptic spiking event occurs (Ros et al., 2006; Roy et al., 2019)). These unique features greatly hinder the running efficiency of brain dynamics models if conventional dense array operators are used (see Section 5.1). Traditional brain simulators utilize a specific data structure and accompanying event-driven computation code to address this problem (Kunkel et al., 2012; Jordan et al., 2018). Nevertheless, this solution is confined to predetermined simulation scenarios, restricting its applicability to other domains. Moreover, it lacks compatibility with automatic differentiation, a crucial element of the backpropagation algorithm, thereby impeding efficient training of brain dynamics models using gradient-based optimization algorithms.

To unify brain simulation and BIC programming workflows, BrainPy abstracts these specialized functions in brain simulation as reusable primitive operators, so that they can be flexibly fused, chained, or combined to define any complex computations of brain models as desired by the user. Particularly, BrainPy provides a set of sparse and event-driven operators in its `brainpy.math.sparse` and `brainpy.math.event` modules, respectively. Sparse operators can not only store synaptic connectivity compactly by avoiding the memory usage of unnecessary zeros, but also compute synaptic currents efficiently by using only nonzero entries. Although modern numerical computing libraries have provided diverse routines for sparse computation, they are prone to encountering problems such as duplicate memory storage of identical synaptic projections and redundant memory access of homogeneous synaptic weights. In contrast, BrainPy's sparse operators prioritize the optimization of sparse computations specifically for brain dynamics modeling. As a result, they offer superior memory efficiency and significantly faster speeds compared to sparse computation routines designed for general domains. Moreover, event-driven operators in BrainPy can further take advantage of the discontinuous nature of spikes by only performing computations when spiking events arrive. They can lead to significant improvements in efficiency (typically orders of magnitude faster, see Section 5.1), as it does not constantly update the state of the system when no spike arrives.

In comparison to traditional brain simulators, our approach of encapsulating the characteristic brain operations as primitive operators streamlines the support of gradient-based optimization algorithms — which are typically used in training BIC models nowadays. Notably, all dedicated operators in BrainPy offer general implementations for both forward- and reverse-mode automatic differentiation, so that brain dynamics models building upon these operators can be differentiable to be used for gradient-based optimization tasks (see Section 5.3).

### 4.2 SCALABLE SIMULATION WITH JIT CONNECTIVITY OPERATORS

Simulating large-scale brain organizations is notoriously challenging since both computing resources and device memory have a near quadratic scaling requirementas the number of neurons increases. For the human brain, which consists of approximately 86 billion neurons, storing the Boolean synaptic connectivity pattern would necessitate nearly 100 terabytes of memory storage. This requirement poses a challenge even for modern supercomputer centers. Moreover, supercomputers are expensive, energy-intensive, and less accessible to a broad range of researchers. This poses significant obstacles for researchers seeking to engage in large-scale brain modeling endeavors.

To address this challenge, BrainPy introduces a JIT connectivity algorithm with extremely low sampling complexity (see Appendix C). JIT connectivity is a method (Lytton et al., 2008; Carvalho et al., 2020; Knight & Nowotny, 2020) used for simulating large-scale brain networks with static synaptic parameters. In this method, synaptic connections are determined by a fixed connectivity rule, and synaptic weights remain unchanged during simulation. Instead of storing the synaptic connectivity, the JIT connectivity algorithm regenerates connectivity parameters when a presynaptic

neuron fires. BrainPy provides a comprehensive range of JIT connectivity operators crafted for performing matrix-matrix multiplication and matrix-vector multiplication within the `brainpy.math.jitconn` module. Compared to standard dense or sparse matrix multiplication operators, BrainPy's JIT operators enable simulations with neural networks two to three orders of magnitude larger on a single device (Section 4.2), paving an easy avenue for large-scale brain simulation for researchers.

## 4.3 INTEGRATING MODELS IN BRAIN SIMULATION AND AI BY DECOUPLING THE DYNAMICS FROM THE COMMUNICATION

Encapsulating characteristic brain operations as primitive operators is an initial step toward integrating brain simulation and AI models. To further this integration, we need a unified perspective for linking major DL components with brain simulation models. The main obstacle hindering this perspective is managing the complex nature of synaptic computation.

BrainPy introduces a unique abstraction that effectively captures and simplifies the complexity of synaptic computations. In BrainPy, a synapse projection between two neuron populations is decomposed into four key components, each encompassing various implementation variants. They are (1) synaptic dynamics, such as alpha, exponential, or dual exponential dynamics; (2) synaptic connectivity patterns, including dense or sparse connections; (3) synaptic output models, such as conductance-based, current-based, or magnesium blocking effects; and (4) synaptic plasticity rules, including short-term plasticity or long-term spike timing dependent plasticity. This decomposition enables BrainPy to offer general implementations for executing diverse synaptic computations. Among the various implementations, two special cases called `AlignPre` and `AlignPost` projections offer superior advantages. Both projections assume homogeneous parameters governing synaptic dynamics within a projection[1]. It is important to note that these projections are not approximations. They accurately compute the same dynamics as the original projections while providing new benefits.

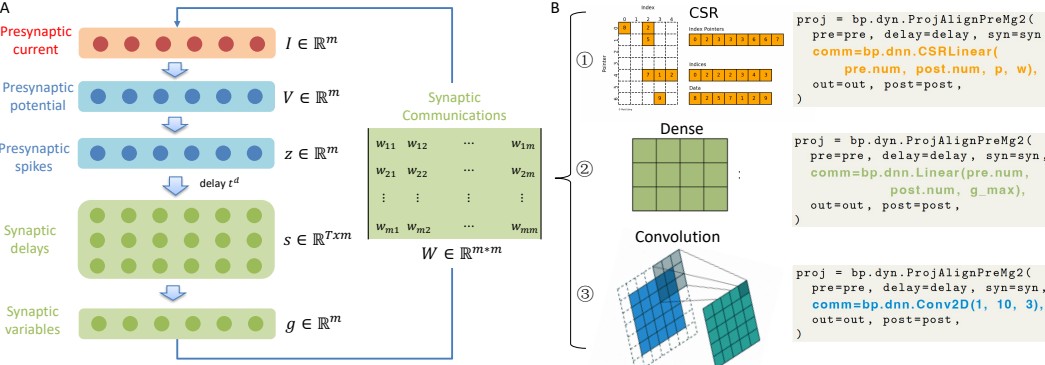

Figure 2: Synaptic projections in BrainPy. (A) The `AlignPre` and `AlignPost` projections offer a decoupled interface for managing dynamics and the communication between dynamics. (B) The synaptic communication allows for diverse implementations, including the utilization of DL models.

**Minimal device memory consumption.** In general, a projection requires storing and updating $pmn$ synaptic variables, where $m$ and $n$ are the numbers of pre- and post-synaptic neurons, and $p$ is the connection probability. However, `AlignPre` and `AlignPost` projections only require $m$ and $n$ variables, respectively, aligning with the dimensions of the pre- and post-synaptic neuron groups (please refer to Appendix D for the reduction details and Figure S7 for the computing workflow). Another aspect that showcases the memory efficiency of `AlignPre` and `AlignPost` projections is their ability to automatically merge duplicate synapse variable creation and updating across multiple projections. `AlingPre` is particularly suitable for handling one-to-many connections since it keeps a trace of synaptic dynamics induced by pre-synaptic neurons (Figure S7C). This trace can be shared and reused across multiple post-synaptic groups if the synaptic parameters

---

[1]We utilize the exponential synapse model to exemplify the homogeneity. The dynamics of the model is described by $\dot{g} = -g/\tau + \sum_k \delta(t - t^k)$, where $g$ the conductance, $\tau$ the time constant, $t_k^{\text{sp}}$ the pre-synaptic spike. We consider the projection homogeneous if the value of $\tau$ remains consistent across all synapses.

(typically, time constants) are the same across these projections. On the other hand, the `AlignPost` projection is highly advantageous for many-to-one connections (Figure S7D). This is because all synaptic interactions with identical time constants can be converged into a single trace of variables for modeling the postsynaptic currents. We showcase the advantages of this automatic synaptic merging by constructing a large-scale spiking network model consisting of 30 brain areas, inspired by the work of Joglekar et al. (2017). Our findings reveal that this technique not only decreases device memory usage but also significantly reduces compilation and simulation time (see Appendix F).

**Decoupling brain dynamics from its communication.** Furthermore, the `AlignPre` and `AlignPost` projections offer a novel perspective on the incorporation of conventional DL components within brain simulation models. As depicted in Figure 2A, there is a distinct separation between the dynamics and the communication between these dynamics. The model dynamics align precisely with the dimensional of pre- and post-synaptic neuron groups, exhibiting strong element-wise and memory-intensive properties. The communication between pre- and post-synaptic groups is facilitated by a communication matrix. Standard brain models implement such communication via sparse matrices, while DL models like linear transformations, convolutions, and normalizations can also serve as alternative communication mechanisms for propagating brain dynamics (Figure 2B).

## 4.4 FLEXIBLE MODELING WITH A MULTI-SCALE MODEL BUILDING INTERFACE

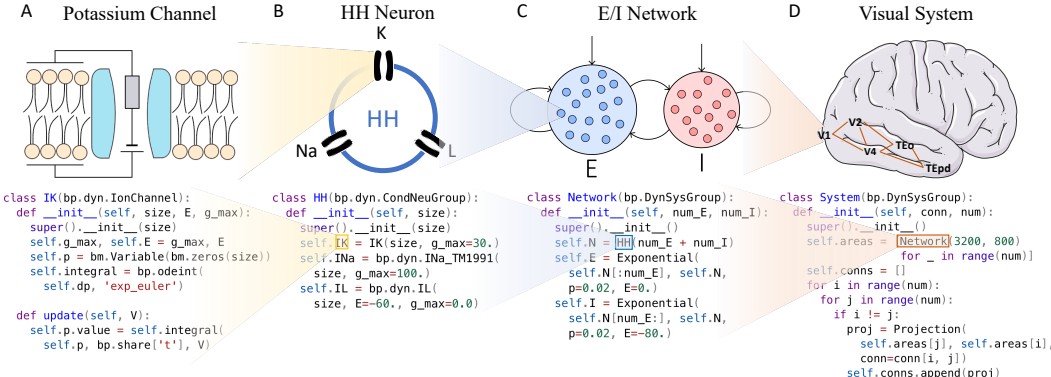

Figure 3: Multi-scale model building interface of BrainPy. Here `bp` is referred to `brainpy` package.

Brain dynamics models have key features of modularity, multi-scale organization, and hierarchy (Meunier et al., 2010). To match these characteristics, BrainPy implements a modular, composable, and flexible programming interface centered around the `brainpy.DynamicalSystem` class. The key idea underlying this multi-scale model-building paradigm is that models at any level of resolution can be defined as `brainpy.DynamicalSystem` classes, and higher-level models (e.g., networks or systems) can be constructed by hierarchically combining lower-level models (e.g., ion channels or neurons). Figure 3 presents an illustrative example of hierarchical model composition. It demonstrates the recursive construction of a model, progressing from channels (Figure 3A) to neurons (Figure 3B), networks (Figure 3C), and finally systems (Figure 3D). It is important to note that while tools like NetPyNE (Dura-Bernal et al., 2019) also enable hierarchical composition, BrainPy clearly distinguishes it through its unique flexibility and customizability. Specifically, BrainPy allows users to flexibly control the depth of composition based on their specific requirements, ensuring seamless alignment with the aforementioned brain characteristics. Additionally, for user convenience, BrainPy offers a wide range of commonly used models such as channels, neurons, synapses, populations, and networks, serving as building blocks to simplify the construction of large-scale models.

## 4.5 OBJECT-ORIENTED JIT COMPILATION

Brain dynamics models are intrinsically memory-intensive. For example, the computation within the classical leaky integrate-and-fire (LIF) neuron primarily consists of element-wise operations such as addition, multiplication, and division. In contrast to DNN models that are typically filled with computation-intensive operators, the memory-intensive nature of brain dynamics models poses

significant challenges for efficient simulation using native Python. To overcome this, BrainPy heavily leverages JAX (Frostig et al., 2018) and XLA (Artemev et al., 2022) for JIT compilation. JIT compilation executes models outside of the Python interpreter and optimizes memory-intensive operators by automatic kernel fusion. This fusion technique effectively reduces off-chip memory access, kernel launching overhead, and CPU-GPU switching delays, making XLA an ideal choice for compiling brain dynamics models. Particularly, BrainPy integrates a collection of object-oriented JIT transformations into its multi-scale model-building interface. These transformations are built upon JAX's implementation of a function-oriented JIT interface, as detailed in Appendix G. By leveraging these transformations, any multi-scale BrainPy model can be effortlessly converted into an optimized XLA process, compatible with CPU, GPU, and TPU platforms.

## 5 DEMONSTRATIONS

In this section, we showcase BrainPy's efficiency and scalability in brain simulation and BIC tasks. We also highlight its differentiable simulation capability by training a biologically plausible spiking network model on working memory tasks.

### 5.1 EFFICIENCY OF BRAINPY

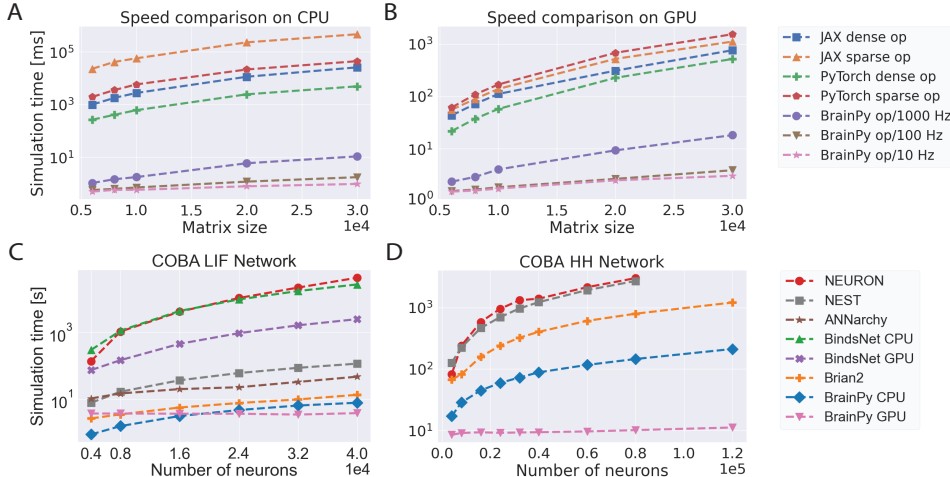

Figure 4: Event-driven operators in BrainPy enable the efficient running of brain simulation models. (A, B) Speed comparison between different operators that perform the matrix-vector multiplication for synaptic computation on both CPU (A) and GPU (B) devices. (C, D) Speed comparison of different brain simulators when simulating the COBA-LIF (C) and COBA-HH networks (D).

We first showcase the performance of our event-driven operators (refer to Section 4.1) by using `brainpy.math.event.csrmv` operator, which is used to compute $\mathbf{y} = \mathbf{M}\mathbf{v}$, where $\mathbf{M} \in \mathbb{R}^{m*n}$ is the synaptic connectivity, $\mathbf{v} \in \mathbb{R}^n$ the presynaptic spikes, and $\mathbf{y} \in \mathbb{R}^m$ the postsynaptic currents. Unlike the alternative dense matrix-vector multiplication, it takes advantage of the CSR representation to store $\mathbf{M}$. Different from the corresponding sparse operator, it makes full use of the event property of the $\mathbf{v}$ and computes only at positions corresponding to the spiking event (see Appendix B). Therefore, `brainpy.math.event.csrmv` is capable of significantly reducing temporal and spatial costs associated with synaptic computations. In practice, we compare `brainpy.math.event .csrmv` with the corresponding sparse and dense operators in JAX and PyTorch. Each operator performs the synaptic computation in 1 s (10,000 times with the time step 0.1 ms), where we randomly sample spiking events in $\mathbf{v}$ so that presynaptic neurons can fire with frequencies at 10 Hz, 100 Hz, and 1000 Hz. The comparison results show that the event-driven operator achieves two to five orders of magnitude faster than other operators on both CPU and GPU devices (Figure 4A and Figure 4B). Moreover, with the lower firing frequency, the event-driven operator runs faster. This is in contrast to its counterparts whose computing times are independent of the number of incoming spiking events.

To verify the efficiency of BrainPy on realistic brain simulation models, we further compare BrainPy with several mainstream brain simulation frameworks, including NEURON (Hines & Carnevale, 1997), NEST (Gewaltig & Diesmann, 2007), Brian2 (Stimberg et al., 2019), ANNArchy (Vitay et al., 2015), and BindsNet (Hazan et al., 2018). We use EI balance networks with the LIF neuron (i.e., the COBA-LIF model (Vogels & Abbott, 2005)) and the Hodgkin-Huxley neuron (i.e., the COBA-HH model (Brette et al., 2007)) as benchmarks. The E/I balanced network typically exhibits sparse and event-driven properties, and is highly suitable to apply `brainpy.math.event.csrmv` for its computation. As shown in Figure 4C and Figure 4D, BrainPy shows the best performance on the COBA-LIF model, and such speed superiority becomes more pronounced in the COBA-HH network.

## 5.2 SCALABILITY OF BRAINPY

We evaluate the scalability of BrainPy by focusing on our JIT connectivity operators (refer to Section 4.2). We investigate the memory usage and execution speed of dense, sparse, and JIT connectivity operator `brainpy.math.jitconn.mv_prob_uniform` when performing the matrix-vector multiplication $\mathbf{y} = \mathbf{J}\mathbf{v}$. Here, $\mathbf{J}$ represents the connection matrix with a connectivity probability of $p$, and the weights at nonzero positions are sampled from a normal distribution $N(\mu, \sigma^2)$ using the seed $s$. $\mathbf{J}$ is stored as a dense matrix in the dense operator, a CSR sparse matrix in the sparse operator, and four scalars $(p, \mu, \sigma, s)$ in the JIT connectivity operator. Our experiments reveal that as the matrix size increases, the JIT connectivity operator maintains nearly constant memory consumption (Figure 5A) and executes with one to two orders of magnitude greater speed compared to the corresponding dense and sparse operators (Figure 5B). This emphasizes the potential for performing large-scale brain simulations on limited computing devices.

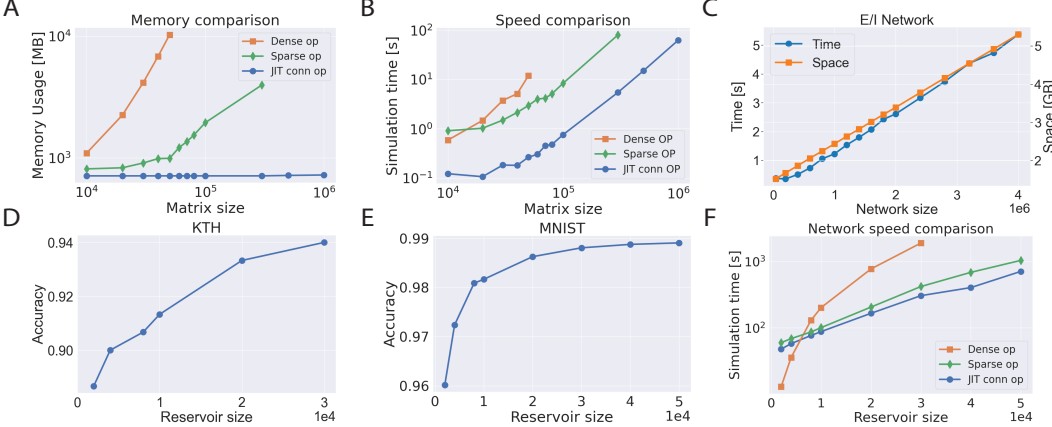

Figure 5: JIT connectivity operators enable large-scale brain dynamics modeling. (A, B) The memory usage (A) and speed (B) comparison between BrainPy's JIT connectivity operator with JAX's sparse and dense operators for performing the matrix multiplication. (C) Scaling up the COBA-LIF network with the JIT connectivity operator. (D, E) The empirical relationship between the classification performance and the reservoir size using KTH (D) and MNIST (E) datasets. (F) The inference speed comparison of the reservoir model using the dense, sparse, and JIT connectivity operators.

To demonstrate its utility in realistic brain simulation, we implement a large-scale version of the aforementioned COBA-LIF network (Vogels & Abbott, 2005) using the proposed JIT connectivity operator (Appendix N). We have successfully scaled up this EI balance network model to over 4 million neurons and 80 incoming synapses per neuron on a single GPU device. The memory and computing time scale linearly with network size (Figure 5C, Figure S16B, and Figure S16C).

Furthermore, these large-scale operators can be applied in brain-inspired reservoir models (Lukoševičius, 2012) as their input and recurrent weights are fixed during training, which aligns well with the capabilities of JIT connectivity operators. To assess its performance, we conducted experiments to scale up the reservoir size (see Appendix H). Initially, we evaluate the model on the KTH dataset (Antonik et al., 2019), and find that the model's performance improves constantly with an increase of the reservoir size (Figure 5D). In a previous study (Antonik et al., 2019), a reservoir with only 16,384 hidden units achieved a testing accuracy of 91.3%. In contrast, our implementation with JIT

connectivity operators allows easy scaling up to a reservoir with 30,000 nodes, resulting in a superior accuracy of up to 94.4%. We also verified this scaling experiment using the MNIST dataset. When the reservoir layer size was set to 50,000 nodes, the network achieved an accuracy of 98.9% (Figure 5E), on par with the state-of-art machine learning algorithms. Additionally, the reservoir model using the JIT connectivity operator is twice as fast during inference compared to the sparse implementation. It also performs better than the dense implementation when the reservoir model has over 10,000 nodes (Figure 5F).

### 5.3 TRAINING FUNCTIONAL BRAIN DYNAMICS MODELS

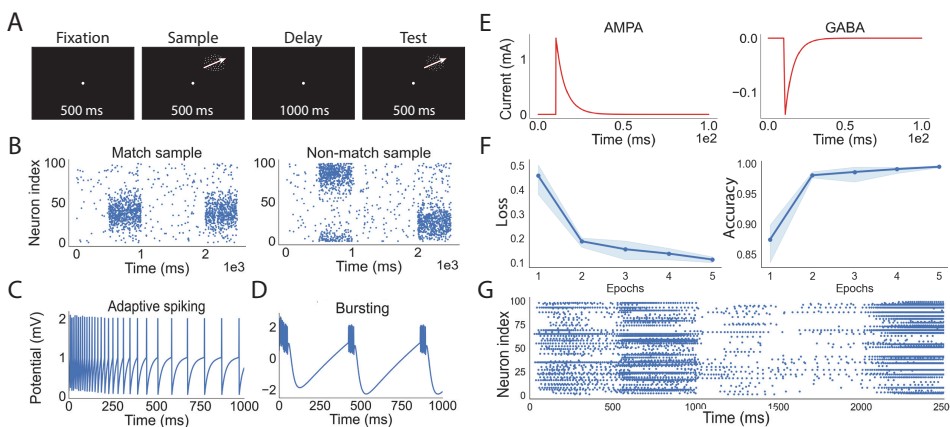

Figure 6: BrainPy helps to train a data-driven, biologically realistic SNN model for performing a working memory task. (A) The delayed match-to-sample working memory task. (B) Two examples that show the *match* case (the motion orientations are consistent between sample and test periods) and the *non-match* case. (C, D) Neuron firing patterns in the model. (E) The AMPA and GABA synapse dynamics in the model. (F) The loss and accuracy of the model when training on the DMS task. (G) An example of the spiking dynamics of the network after training.

Finally, we evaluate the differentiable simulation capability of BrainPy for biological brain networks. Particularly, we apply the back-propagation algorithm to train a data-driven SNN model of the prefrontal cortex (PFC) (refer to Appendix I) with a working memory task (Figure 6A and Figure 6B). We use the generalized leaky integrate-and-fire (GIF) model to fit the spiking patterns of PFC neurons as observed in experiments (Mihalas & Niebur, 2009). After fitting, most PFC neurons exhibit tonic spiking with spike frequency adaptation (Figure 6C), while the remaining have characteristic bursting features (Figure 6D). Furthermore, we consider the exponential synapse model to model AMPA and GABA synapses (Figure 6E). We train the network to solve the delayed match-to-sample (DMS) task, where the network must indicate whether sequentially presented sample and test stimuli (500 ms sample period; 1000 ms delay period) match exactly (Figure 6A). In this task, the network must maintain the stimulus information for an extended time period (> 1000 ms), then compare the held information to the test stimulus to make a decision. We find that our biologically grounded GIF network can successfully solve this task requiring long-term dependencies. Within a few epochs, the testing accuracy rapidly increases to nearly 100% (Figure 6F). The post-training spiking dynamics of the network also exhibit comparable patterns to the neural activity of PFC neurons recorded from monkeys performing the same DMS task (see Figure 6G and data in Constantinidis et al. (2016)).

## 6 CONCLUSION

In conclusion, BrainPy provides a *differential brain simulator* that serves as a bridge between the worlds of *brain simulators* and *DL frameworks*. By leveraging dedicated optimizations, it enables flexible, efficient, and scalable brain simulation capabilities within the JAX framework. Additionally, its seamless integration with the JAX machine learning ecosystem facilitates the incorporation of AI models into brain simulation. Through the unique combination of these two strengths, BrainPy emerges as a powerful tool for exploring the complexities of the brain and a comprehensive platform for interdisciplinary research between brain simulation and BIC.

## REPRODUCIBILITY STATEMENT

We put all the codes in our supplementary materials for reproducing our experiment results which are mainly demonstrated in Section 5. Other results in the appendix are also included in our codes. We provide a README file for our code directory structure and running guidance. For the details of Section 5.2 and Section 5.3 , a complete description of the model and training parameters are given in Appendix H and Appendix I.

## ACKNOWLEDGMENTS

This work was supported by the Science and Technology Innovation 2030-Brain Science and Brain-inspired Intelligence Project (No. 2021ZD0200204). We thank Yifeng Gong from Beijing Institute of Technology for his valuable contributions during his internship in Si Wu's lab.

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

## A  ENVIRONMENT SETTINGS

In this paper, all evaluations and benchmarks were conducted in a Python 3.10 environment, which was installed on a system running Ubuntu 22.04.2 LTS. For CPU experiments, we used Intel(R) Xeon(R) W-2255 CPU @ 3.70GHz, 64GB RAM @ 3200MHz. For GPU experiments, we used the NVIDIA RTX$^{\text{TM}}$ A6000 GPU with CUDA 11.7.

For brain simulation, we compared the running performance of BrainPy with state-of-art brain simulators, including NEURON (Hines & Carnevale, 1997) in version 8.2.0, NEST (Gewaltig & Diesmann, 2007) at version 3.6, Brian2 (Stimberg et al., 2019) at version 2.5.1, ANNArchy (Vitay et al., 2015) in version 4.7.2, and BindsNet (Hazan et al., 2018) in version 0.3.2,

For brain-inspired computing, we compared BrainPy with several spiking neural networks (SNN) training packages, including snnTorch (Eshraghian et al., 2021) in version 0.6.1, SpikingJelly (Fang et al., 2020) in version 0.0.0.0.14, and Norse (Pehle & Pedersen, 2021) in version 1.0.0.

When comparing the performance of dedicated operators in BrainPy with conventional operators in modern deep learning frameworks, we used PyTorch (Paszke et al., 2019) at version 2.0, and JAX (Frostig et al., 2018) at version 0.4.10.

## B  EVENT-DRIVEN MATRIX-VECTOR MULTIPLICATION

Event-driven matrix-vector multiplication $\mathbf{y} = \mathbf{M}\mathbf{v}$ in BrainPy is used for synaptic computation, where $\mathbf{v}$ is the presynaptic spikes, $\mathbf{M}$ the synaptic connection matrix, and $\mathbf{y}$ the postsynaptic current. Specifically, it performs matrix-vector multiplication in a sparse and efficient way by exploiting the event property of the input vector $\mathbf{v}$. Instead of multiplying the entire matrix $\mathbf{M}$ by the vector $\mathbf{v}$, which can be wasteful if $\mathbf{v}$ has many zero elements, event-driven matrix-vector multiplication in BrainPy only performs multiplications for the non-zero elements of the vector, which are called events. This can reduce the number of operations and memory accesses, and improve the running performance of matrix-vector multiplication.

In particular, BrainPy implements `brainpy.math.event.csrmv`, in which the connection matrix $\mathbf{M}$ is stored as the compressed sparse row (CSR) sparse matrix. The computation of this operator requires several parameters:

1. `data`: The weights of $\mathbf{M}$ contain the non-zero elements in the row-major order.
2. `indices`: The array contains the column indices of the non-zero elements in the matrix $\mathbf{M}$.
3. `indptr`: The array contains the starting index of each row.
4. `events`: The presynaptic spiking vector $\mathbf{v}$.
5. `shape`: A tuple with two integers representing the shape of the matrix $\mathbf{M}$.

`brainpy.math.event.csrmv` makes full use of the event property of $\mathbf{v}$, and computes only at positions where the spike in $\mathbf{v}$ is `True`. The pseudo-code of this operator can be written in Listing S1.

```python
def csrmv(data, indices, idnptr, events, outs):
  for i, event in enumerate(events):
    if event:
      for j in range(idnptr[i], idnptr[i + 1]):
        outs[indices[i]] += data[j]
```

Listing S1: The Python pseudo-code of `brainpy.math.event.csrmv`, where `data`, `indices`, and `idnptr` corresponds to the matrix $\mathbf{M}$, `events` indicates the vector $\mathbf{v}$, and `outs` represents the postsynaptic current $\mathbf{y}$.

## C  MATRIX-VECTOR MULTIPLICATION WITH THE JUST-IN-TIME CONNECTIVITY

Synaptic connectivity storage imposes a memory bottleneck for large-scale neuronal network simulations, as it scales quadratically with the number of neurons. Previous studies have demonstrated that

static synaptic parameters in brain modeling can be dynamically regenerated during runtime, thereby circumventing the space costs of connectivity Lytton et al. (2008); Carvalho et al. (2020); Knight & Nowotny (2020).

We investigate the dynamic regeneration of synaptic connectivity information in a matrix-vector product operation $\mathbf{y} = \mathbf{Mv}$, where $\mathbf{M}$ is the synaptic connection matrix to be regenerated, $\mathbf{v}$ is the presynaptic spike vector, and $\mathbf{y}$ is the postsynaptic current vector.

A common connectivity schema is the fixed probability connection, where each neuron in the presynaptic population connects to each neuron in the postsynaptic population with a fixed probability $p$. The postsynaptic targets of any presynaptic neuron can thus be drawn from a Bernoulli process with success probability $p$.

A naive way of drawing from the Bernoulli process is to sample repeatedly from the uniform distribution $U[0, 1]$ and create a synapse if the sample is smaller than $p$. However, this is highly inefficient for sparse connectivity ($p \ll 1$).

A more efficient way of drawing the sampling is using the geometric distribution $\text{Geo}[p]$ (Knight & Nowotny, 2020). Instead of sampling at every possible connection position, we can sample the distance between two consecutive connection positions, avoiding unnecessary sampling at failed positions. The geometric distribution can be sampled in constant time by inverting the cumulative density function of the corresponding continuous distribution to obtain $\frac{\log(U[0,1])}{\log(1-P)}$. However, this sampling method is still costly on CPU and GPU devices due to the existence of `log` operation.

We propose to sample the distance between two consecutive connection positions using the uniform random integers from 1 to $\lfloor \frac{2}{p} - 1 \rfloor$. The expectation of $U[1, \lfloor \frac{2}{p} - 1 \rfloor]$ is $\frac{1}{p}$, ensuring that the sampling has the desired connection probability of $p$. In practice, sampling from $U[1, \lfloor \frac{2}{p} - 1 \rfloor]$ is one order of magnitude faster than sampling from $\text{Geo}[p]$. The corresponding Python pseudo-code of our proposed solution is shown in Listing S2.

```python
import math, random

def jitconn_prob_homo(events, prob, weight, seed, outs):
  random.seed(seed)
  max_cdist = math.ceil(2/prob - 1)
  for event in events:
    if event:
      post_i = random.randint(1, max_cdist)
      while post_i < len(outs):
        outs[post_i] += weight
        post_i += random.randint(1, max_cdist)
```

Listing S2: The Python pseudo-code of our just-in-time connectivity operator, where `events` indicates the presynaptic spikes $\mathbf{v}$, `prob`, `weight`, and `seed` corresponds to the matrix $\mathbf{M}$, and `outs` represents the postsynaptic current $\mathbf{y}$. The `seed` parameter guarantees the reproducibility and consistency of matrix regeneration across multiple invocations of this function. Note here, all nonzero elements in the matrix $\mathbf{M}$ have the save value `weight`.

## D  SYNAPTIC PROJECTIONS WITH ALIGNPRE AND ALIGNPOST

`AlignPre` holds true because with homogeneous synaptic parameters, the spike train coming from the same presynaptic neuron will lead to the same synaptic dynamics. As a result, all synapses originating from the same presynaptic neuron can share a single dynamical variable. Moreover, the `AlignPre` projection is suitable for all types of synapse models.

In contrast, the applicability of `AlignPost` is limited to synapse models that exhibit exponential dynamics or are composed of exponential components. For the exponential synapse, the conductance $g$ of a post-synaptic neuron is updated according to $g \leftarrow g + 1$ whenever a spike arrives and regardless of which presynaptic neuron emitted this spike (refer to Appendix E for details).

`AlignPre` is capable of modeling scenarios where the pre-synaptic neuron group projects to multiple downstream post-synaptic neuron groups. This mechanism imposes a constraint on the size of synapse

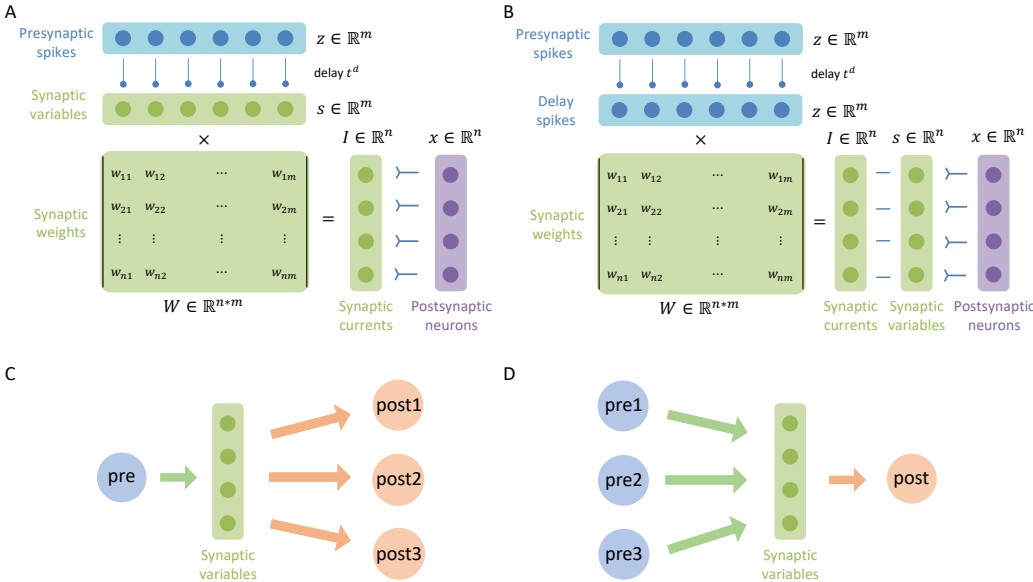

Figure S7: Overview of Synaptic Projections. (A) The workflow of the `AlignPre` model. (B) The workflow of the `AlignPost` model. (C) The `AlignPre` model is suitable for one-to-many connection. (D) The `AlignPost` model is suitable for many-to-one connection.

variables, ensuring alignment with the number of pre-synaptic neuron groups. Consequently, if these projections share the same delay, only a single synapse variable is stored for all these projections. The operational sequence of `AlignPre` unfolds as follows: Initially, it receives spikes from the pre-synaptic neuron group. By considering the delay times, it retrieves spikes that occur after the specified delay and then proceeds to compute synapse variables through synaptic dynamics. Subsequently, the synapse variables traverse the synaptic communication layer and synaptic output layer, where they are transformed into synaptic currents, aligned in accordance with the number of post-synaptic neuron groups. These synaptic currents are then transmitted to the post-synaptic neuron groups for further computations.

`AlignPost`, on the other hand, is designed to handle scenarios in which the post-synaptic neuron group receives input from multiple upstream pre-synaptic neuron groups. In this case, the size of the synaptic variables is adjusted to match the number of post-synaptic neuron groups. The operational sequence of `AlignPost` closely resembles that of `AlignPre`, with the notable difference being that the calculation of synaptic variables occurs after the synaptic communication layer. Specifically, `AlignPost` combines delayed pre-synaptic spikes with synaptic weights to update synaptic variables. Subsequently, these updated synaptic variables undergo computation through the synaptic output layer, resulting in synaptic currents that are then conveyed to the post-synaptic neurons.

## E    EXPONENTIAL SYNAPSE MODEL ENABLES THE `AlignPost` PROJECTION

For the exponential synapse, the conductance $g$ of a post-synaptic neuron is governed by

$$g(t) = \sum_{j}^{n} \exp\left(-\frac{t - t_j^{sp}}{\tau}\right), \tag{1}$$

where $n$ is the total number of spikes the post-synaptic neuron received at the current time $t$, $t_j^{sp}$ the spiking time of the received $j$-th spike, and $\tau$ the synaptic time constant.

Eq (1) can be rewritten as a differential equation:

$$\dot{g} = -\frac{g(t)}{\tau} + \sum_{i,k} \delta(t - t_{i,k}^{sp}), \tag{2}$$

where $i$ is the index of the connected pre-synaptic neuron, and $t^{\text{sp}}_{i,k}$ is the time of $k$-th spike of the pre-synaptic neuron $i$.

In the discrete version, Eq (2) is equivalent to the following equations:

$$g(t) = \exp(-\Delta t/\tau)g(t - \Delta t), \tag{3}$$

and whenever a pre-synaptic spike arrives, $g(t)$ undergoes an update according to:

$$g(t) \leftarrow g(t) + 1. \tag{4}$$

Therefore, the exponential dynamics property enables us to track and record information using a single variable for each post-synaptic neuron.

## F    MULTI-AREA SPIKING NEURAL NETWORK

We build a spiking network model which is inspired by Joglekar et al. (2017) with 30 brain areas. Simulations are performed using a network of leaky integrate-and-fire neurons, with the local circuit and long-range connectivity structure derived from (Markov et al., 2014). Each of the five areas consists of 4000 neurons, with 3200 excitatory and 800 inhibitory neurons.

For each neuron, the membrane potential dynamics are described by the following equations:

$$\tau \frac{dV}{dt} = -(V(t) - V_{rest}) + RG(t) \tag{5}$$

when $V(t) > V_{th}$, the neuron generates a spike and $V(t)$ is set to $V_{reset}$. $V$ is the membrane potential, $V_{rest}$ is the resting membrane potential, $V_{reset}$ is the reset membrane potential, $V_{th}$ is the spike threshold, $\tau$ is the time constant, $R = 1\,\Omega$ is the resistance, and $G(t)$ is the time-variant synaptic inputs.

For the synapse, we use conductance-based synaptic interactions. Particularly, $G(t)$ is given by:

$$G(t) = -\sum_j g_{ji}(t)\,(V_i - E_j)\,, \tag{6}$$

where $V_i$ is the membrane potential of neuron $i$. The synaptic conductance from neuron $j$ to neuron $i$ is represented by $g_{ji}(t)$, while $E_j$ signifies the reversal potential of that synapse. For excitatory synapses, $E_j$ was set to 0 mV, whereas for inhibitory synapses, it was -80 mV.

The dynamics of the synaptic conductance is given by:

$$\frac{dg_{ji}}{dt} = -\frac{g_{ji}}{\tau_{\text{decay}}} + g_{\max} \sum_k \delta(t - t^k_j), \tag{7}$$

where $t^k_j$ is the spiking time of the presynaptic spike. Whenever a spike occurred in neuron $j$, the synaptic conductance $g_{ji}$ experienced an immediate increase by a fixed amount $g_{\max}$. Subsequently, the conductance $g_{ji}$ decayed exponentially with a time constant of $\tau_{\text{decay}} = 5$ ms for excitation and $\tau_{\text{decay}} = 10$ ms for inhibition.

The connection density is set according to the experimental connectivity data (Markov et al., 2014). The inter-areal connectivity is measured as a weight index, called the extrinsic fraction of labeled neurons (Markov et al., 2014). The intra-area connectivity is set according to the setting in a standard EI balance network (Brette et al., 2007).

Moreover, we introduce distance-dependent inter-areal synaptic delays by assuming a conduction velocity of 3.5 m/sec (Swadlow, 1990) and using a distance matrix based on experimentally measured wiring distances across areas (Markov et al., 2014).

We build this multi-area model using the synaptic projection with and without `AlignPost`. Then we measure the memory usage after constructing the model, the compilation time during JIT compilation, and the execution time when simulation the model. The experimental data is avaliable in Figure S8.

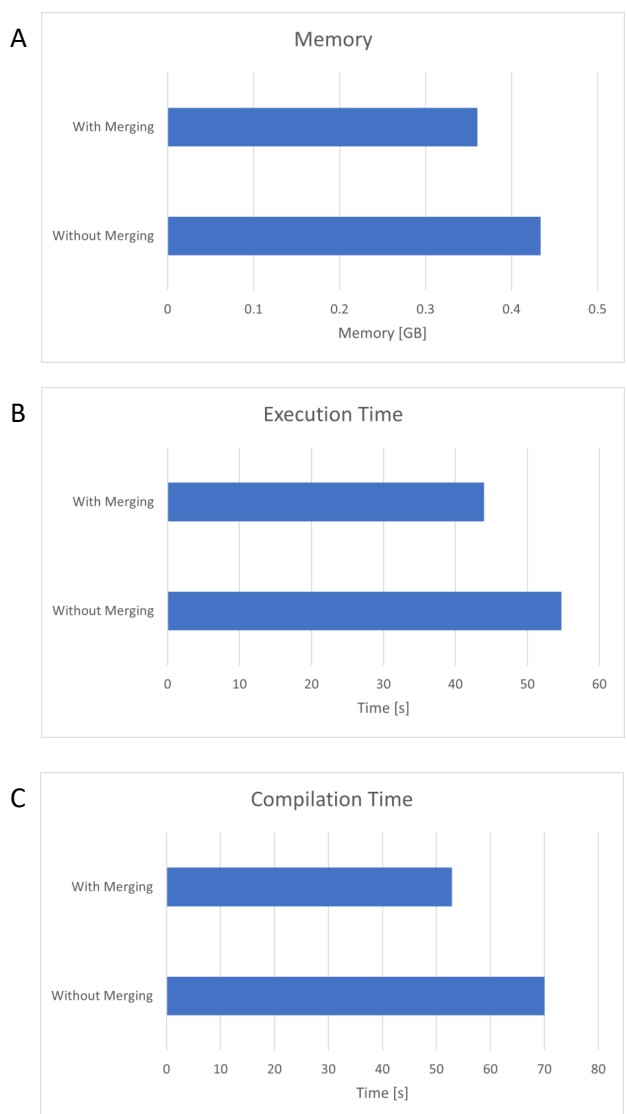

Figure S8: The evaluation of memory usage, model compilation time, and model execution time for the multi-are spiking neural network model using synaptic projection with and without the automatic merging of the `AlignPost` projection.

## G  OO TRANSFORMATION COMPARISON

The JIT compilation interface of JAX/XLA is designed to work with pure Python functions, which is not compatible with the modular and composable programming interface in BrainPy for building brain dynamics models (see Section 4.4). To bridge this gap, we provide `brainpy.math.jit` for automatic object-oriented JIT transformation.

In this object-oriented transformation, variables that will change during execution should be declared as `brainpy.math.Variable` (otherwise, they will be treated as constants and compiled into the transformed models). Then, a single line calling `model_transformed = brainpy.math.jit (model)` will transform the `model` object into a function compatible with JAX's functional JIT interface, before final compilation into an optimized XLA process `model_transformed`. Through this streamlined approach, BrainPy's object-oriented JIT compilation furnishes a robust infrastructure to maximize the performance of diverse brain dynamics models.

Although Flax and Haiku provide functionality for managing state variables, they do not focus specifically on the needs of brain dynamics modeling. In Flax, the initialization of state variables is mixed into the computation, making it hard to separate variables from computing logic. In Haiku, state variables have a complex syntax for declaration and calculation, which increases the difficulties for users to use these time-dependent variables in complex brain dynamical systems.

The following codes compares the OO programming of Haiku, Flax, and BrainPy, and it highlight how straightforward the BrainPy is:

```python
import haiku as hk
import jax.numpy as jnp
import jax.random as jr

def stateful_f():
  counter = hk.get_state("counter", shape=[], dtype=jnp.int32, init=jnp.
    ones)
  hk.set_state("counter", counter + 1)

f = hk.without_apply_rng(hk.transform_with_state(stateful_f))
params, state = f.init(rng=jr.PRNGKey(1))
for i in range(3):
  _, state = f.apply(params, state)
  print(f'After {i + 1} iterations, State: {state}')
```
Listing S3: The Haiku's object-oriented JIT interface.

```python
from flax import linen as nn
import jax.numpy as jnp
import jax.random as jr

class F(nn.Module):
  @nn.compact
  def __call__(self):
    counter = self.variable('state', 'counter', lambda s: jnp.ones(s, jnp
    .int32), ())
    counter.value += 1

f = F()
variables = f.init(jr.PRNGKey(1))
for i in range(3):
  _, variables = f.apply(variables, mutable=['state'])
  print(f'After {i + 1} iterations, State: {dict(variables["state"])}')
```
Listing S4: The Flax's object-oriented JIT interface.

```python
import brainpy as bp
import brainpy.math as bm

class F:
  def __init__(self):
    self.counter = bm.Variable(bm.ones((), dtype=bm.int32))

  @bm.cls_jit
  def __call__(self):
    self.counter += 1

f = F()
for i in range(3):
  _ = f()
  print(f'After {i + 1} iterations, State: {f.counter}')
```
Listing S5: The BrainPy's object-oriented JIT interface.

The Pythonic approach of BrainPy makes the object-oriented JIT much more accessible for neuroscientists without deep ML expertise and lowers the barriers compared to Flax or Haiku.

## H  THE RESERVOIR COMPUTING MODEL

**Reservoir model.**  The dynamics of the reservoir model used here is given by (Lukoševičius, 2012):

$$\mathbf{x}(t) = (1 - \alpha) \cdot \mathbf{x}(t - 1) + \alpha \cdot f(\mathbf{W}_{in}\,\mathbf{u}(t) + \mathbf{W}_{rec}\,\mathbf{x}(t - 1)), \qquad (8)$$

$$\mathbf{y}(t) = \mathbf{W}_{out}\,\mathbf{x}(t) \qquad (9)$$

where $\mathbf{x}(t)$ is the reservoir state, $\mathbf{y}(t)$ the readout value, $\mathbf{W}_{in}$ and $\mathbf{W}_{rec}$ are input and recurrent connection matrices, respectively, $\mathbf{W}_{out}$ the readout weight matrix which can be trained by either offline learning or online learning algorithms, $\alpha \in (0, 1]$ the leaky rate, $\mathbf{u}(t)$ the input at time step $t$, and $f$ the nonlinear activation function.

**Model implementation.**  $\mathbf{W}_{in}$ and $\mathbf{W}_{rec}$ are fixed and randomly initialized, and they are highly suitable to be implemented with the just-in-time connectivity operators.

Since $\mathbf{W}_{in}$ is usually initialized with the uniform distribution $U[-s, s]$, we implement the input operation $\mathbf{W}_{in}\mathbf{u}(t)$ with `brainpy.math.jitconn.mv_prob_uniform(vector, w_low, w_high , conn_prob, seed)`, where `vector` corresponds to the input $\mathbf{u}(t)$, `w_low` corresponds to $-s$, `w_high` corresponds to $s$, `conn_prob` corresponds to the connection probability of the input matrix $\mathbf{W}_{in}$, and `seed` the random seed controlling the reproducibility of the input matrix.

$\mathbf{W}_{rec}$ is usually initialized with the normal distribution $N(0, \sigma)$ with a desirable spectral radius $\rho$. Usually, we have the relationship of $\sigma = \rho/\sqrt{m * p}$, where $p$ is the connection probability of the matrix, $m$ the matrix size, and $\rho$ the spectral radius of the recurrent weight $\mathbf{W}_{rec}$. Therefore, we implement the recurrent matrix operation $\mathbf{W}_{rec}\,\mathbf{x}(t - 1)$ with `brainpy.math. jitconn.mv_prob_normal(vector, w_mu, w_sigma, conn_prob, seed)`, where `vector` corresponds to the reservoir state $\mathbf{x}(t - 1)$, `w_mu` is 0, `w_sigma` corresponds to $\sigma$, `conn_prob` corresponds to the connection probability of the recurrent matrix $\mathbf{W}_{rec}$, and `seed` the random seed controlling the reproducibility of the recurrent matrix.

**Training methods.**  The training objective of reservoir models is to find the optimal $\mathbf{W}_{out}$ that minimizes the square error between $\mathbf{y}(t)$ and $\mathbf{y}^{target}(t)$. The common way to learn the linear output weight $\mathbf{W}_{out}$ is using the ridge regression algorithm (Lukoševičius, 2012). However, ridge regression is an offline learning method in which the parameters are learned given all samples of data. When training reservoir models with a large amount of samples, ridge regression usually is halted by the limited memory space of devices. Therefore, we use the FORCE learning method (Sussillo & Abbott, 2009) to train the model. Contrary to ridge regression, FORCE learning is an online learning algorithm that learns the linear readout weight using only local information in time. This learning mechanism can update the parameters of a model one sample of data at a time, which can significantly reduce computational costs and enable training a large-scale reservoir to be possible.

Table S1: The parameter table of the reservoir model on different datasets.

| Parameter | KTH dataset | MNIST dataset |
|---|---|---|
| $\mathbf{W}_{in}$ connection probability | [0.01, 0.005] | 0.1 |
| $\mathbf{W}_{rec}$ connection probability | [0.001, 0.0002, 0.0001] | [0.1, 0.01] |
| Distribution of $\mathbf{W}_{in}$ | Uniform distribution | Uniform distribution |
| Distribution of $\mathbf{W}_{rec}$ | Normal distribution | Normal distribution |
| Spectral radius $\rho$ | 1.0 | 1.3 |
| Input scaling $s$ | 0.1 | 0.3 |
| Leaky rate $\alpha$ | 0.9 | 0.6 |
| Number of training epoch | 10 | 5 |
| FORCE learning rate | 0.1 | 0.1 |

**Experiment details.**  We conducted several experiments to investigate the performance of large-scale reservoir models on different datasets. First, we evaluate the performance of the model on the KTH dataset (Antonik et al., 2019), a widely used benchmark dataset for action recognition in computer vision research. This dataset contains spatiotemporal patterns that can be captured by

reservoir models. Then we evaluate the reservoir model on the MNIST dataset, which is a static image dataset without temporal information. All parameter selections for classifying two datasets are shown in Table S1.

We set the connection probabilities of $\mathbf{W}_{in}$ and $\mathbf{W}_{rec}$ as follows: For the KTH dataset, $\mathbf{W}_{in}$ has a connection probability of 0.01 for size $\in [2000, 4000, 8000, 10000, 20000]$ and 0.005 for size $\in [30000]$. $\mathbf{W}_{rec}$ has a connection probability of0.001 for size $\in [2000, 4000, 8000]$, 0.0002 for size $\in [10000]$, and 0.0001 for size $\in [20000, 30000]$. For the MNIST dataset, $\mathbf{W}_{in}$ has a connection probability of 0.1 for all reservoir sizes, and $\mathbf{W}_{rec}$ has a connection probability of 0.1 for size $\in [2000, 4000, 8000, 10000]$, and 0.01 for size $\in [20000, 30000, 40000, 50000]$.

## I    RECURRENT NEURAL NETWORKS FOR PERFORMING THE DMS TASK

**Network structure.**    The architecture of recurrent networks used here (both the rate- and spiking-based models) is shown in Figure S9, where the recurrent layer consists of recurrent units that receive and process the time-varying inputs from the input layer, and generate the desired time-varying outputs. The input layer encodes the sensory information relevant to the task, while the readout layer produces a decision in terms of an abstract decision variable.

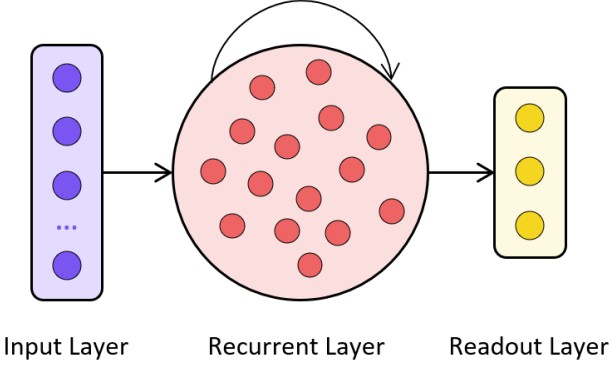

Figure S9: The schematic diagram of the recurrent neural network model for implementing a working memory task.

**The input layer.**    The input layer consists of $N$ motion direction-tuned input neurons (for the spiking network, $N = 100$; for the rate model, $N = 24$). The tuning of the motion direction-selective neurons followed a von Mises distribution, such that the firing rate activity of the input neuron $i$ during the stimulus period was

$$u_t^i = A \exp \left( \kappa \cos \left( \theta - \theta_{\text{pref}}^i \right) \right), \tag{10}$$

where $\theta$ is the direction of the stimulus, $\theta_{\text{pref}}^i$ is the preferred direction of input neuron $i$, $\kappa$ was set to 2.0, and $A$ is the maximum firing rate and was set to 40 Hz. Furthermore, for all periods, input neurons keep a constant background firing rate $r_{\text{bg}}$. Here, $r_{\text{bg}}$ was set to 1 Hz.

In the main text, Figure 3B depicts the spiking activities of the input layer in the spiking-based model. Unlike the spiking encoded input, which produces discrete spikes, the rate-based input layer generates continuous firing rate values as data. Figure S10 demonstrates the cases of *match* (when the stimuli during the sample and test periods are identical) and *non-match* (when they are different).

**The recurrent layer in the rate model.**    In the rate-based model, the recurrent layer consists of 80 excitatory neurons and 20 inhibitory neurons. The firing activity $\mathbf{r}$ of the rate-based recurrent neurons was modeled with the following dynamical equation (Song et al., 2016):

$$\tau \frac{d\boldsymbol{r}}{dt} = -r + f \left( W^{\text{rec}} \boldsymbol{r} + W^{\text{in}} \boldsymbol{u} + \boldsymbol{b}^{\text{rec}} + \sqrt{2\tau} \sigma_{\text{rec}} \zeta \right), \tag{11}$$

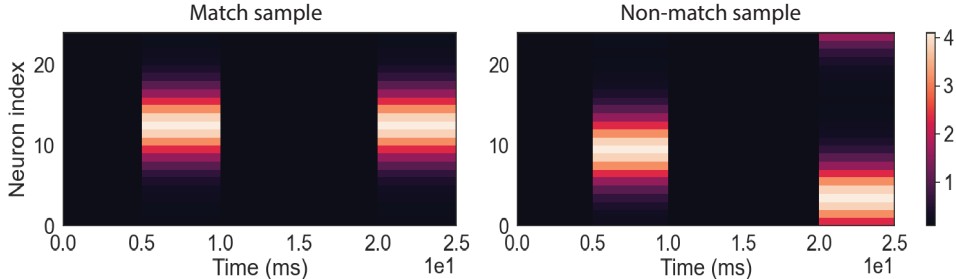

Figure S10: The rate version of two examples that show the *match* case and the *non-match* case. For the spiking version of the input, please refer to Figure 3 in the main text.

where $\tau$ is the neuron time constant, $f(*)$ represents the activation function, $W^{\text{rec}}$ and $W^{\text{in}}$ are the recurrent synaptic weights and input synaptic weights, respectively, $b^{\text{rec}}$ is a bias term, $\zeta$ is independent Gaussian white noise with zero mean and unit variance applied to all recurrent neurons and $\sigma_{\text{rec}}$ is the noise intensity. The activation function used in this study is $f(x) = \max(0, x)$.

In practice, the simulation of the Eq. 11 was performed with the Euler integration method:

$$\boldsymbol{r}_t = (1 - \alpha)\boldsymbol{r}_{t-dt} + \alpha f\left(W^{\text{rec}}\boldsymbol{r}_{t-dt} + W^{\text{in}}\boldsymbol{u}_t + \boldsymbol{b}^{\text{rec}} + \sqrt{\frac{2}{\alpha}}\sigma_{\text{rec}}N(0,1)\right), \tag{12}$$

where $\alpha = dt/\tau$, $dt$ the numerical integration step, $N(0,1)$ denotes the standard normal distribution.

To maintain the separation of excitatory and inhibitory neurons, we decomposed the recurrent weight matrix $W^{\text{rec}}$ as the product between a trainable non-negative matrix $W^{\text{rec},+}$ and a fixed diagonal matrix $D$ (Song et al., 2016):

$$W^{\text{rec}} = W^{\text{rec},+}D \tag{13}$$

$$D = \begin{bmatrix} 1 & & \\ & \ddots & \\ & & -1 \end{bmatrix} \tag{14}$$

**The recurrent layer in the spiking model.** For the spiking model, the recurrent cell was implemented with the spiking neuron model. In this study, the spiking neuron is modified from the generalized integrate-and-fire neuron model (Mihalas & Niebur, 2009). In particular, this model has two internal currents, one fast and one slow. Its dynamic behavior is given by

$$\tau_{I1}\frac{d\mathbf{I_1}}{dt} = -\mathbf{I_1}, \qquad \text{fast internal current} \tag{15}$$

$$\tau_{I2}\frac{d\mathbf{I_2}}{dt} = -\mathbf{I_2}, \qquad \text{slow internal current} \tag{16}$$

$$\tau_V\frac{d\mathbf{V}}{dt} = -\mathbf{V} + V_{rest} + R(\mathbf{I_1} + \mathbf{I_2} + \mathbf{I_{ext}}), \qquad \text{membrane potential} \tag{17}$$

When $V^i$ of $i$-th neuron meets $V_{th}$, the modified GIF model fires:

$$I_1^i \leftarrow A_1, \tag{18}$$

$$I_2^i \leftarrow I_2^i + A_2, \tag{19}$$

$$V^i \leftarrow V_{rest}, \tag{20}$$

where $\tau_{I1}$ denotes the time constant of the fast internal current, $\tau_{I2}$ the time constant of the slow internal current, $\tau_V$ the time constant of membrane potential, $R$ the resistance, $\mathbf{I_{ext}}$ the external input, $V_{rest}$ the resting potential, and $A_1$ and $A_2$ the spike-triggered currents.

For matching the statistical data of electrophysiological experiments in Hass et al. (2016), we assigned 80 excitatory neurons and 20 inhibitory neurons. Inhibitory neurons exhibit a bursting firing pattern,

while excitatory neurons show adaptive spikings. We set $A_1 = 8.0$ for inhibitory neurons, $A_1 = 0$ for excitatory neurons, $A_2 = -0.6$, $\tau_{I1} = 10$ ms, $\tau_{I2}$ was sampled from a uniform distribution $U[100, 3000]$ ms.

The external current $\mathbf{I_{ext}}$ in the network was modeled through the exponential synapse model, whose dynamics is given by:

$$\frac{dI_{ext}^i}{dt} = -\frac{I_{ext}^i}{\tau_{decay}} + \sum_j W_{ij}^{rec} z^j[t - d^{ij}] + \sum_j W_{ij}^{in} u^j[t], \tag{21}$$

where $\tau_{decay}$ is the time constant of the synaptic state, $t^k$ the time of the presynaptic spike, $W^{rec}$ the recurrent weight, $W^{in}$ the input to recurrent weight, and $d^{ij}$ the synaptic delay. Here, $d^{ij} = 0$ ms. $\tau_{decay} = 100$ ms was chosen according to the previous literature (Compte et al., 2000).

To inspect the computational graph of the modified GIF network, we also give the discrete description of the model:

$$\mathbf{I_1}[t + \Delta t] = \text{where}(\mathbf{z}[t] == 1, A_1, \alpha_{I_1} \mathbf{I_1}[t]) \tag{22}$$
$$\mathbf{I_2}[t + \Delta t] = \alpha_{I_2} \mathbf{I_2}[t] + A_2 \mathbf{z}[t] \tag{23}$$
$$\mathbf{I_{ext}}[t + \Delta t] = \alpha_{I_{ext}} \mathbf{I_{ext}}[t] + W_{rec} \mathbf{z}[t] + W_{in} \mathbf{u}[t] \tag{24}$$
$$\mathbf{V}[t + \Delta t] = \alpha_V \mathbf{V}[t] + (V_{rest} + R(\mathbf{I_1}[t + \Delta t] + \mathbf{I_2}[t + \Delta t] + \mathbf{I_{ext}}[t]))\Delta t \tag{25}$$
$$\mathbf{z}[t + \Delta t] = \text{where}(\mathbf{V}[t + \Delta t] > V_{th}, 1, 0) \tag{26}$$

where $z$ is the spiking state, $\alpha_{I_1} = e^{-\frac{1}{\tau_{I_1}}\Delta t}$, $\alpha_{I_2} = e^{-\frac{1}{\tau_{I_2}}\Delta t}$, $\alpha_{V_{th}} = e^{-\frac{1}{\tau_{V_{th}}}\Delta t}$, $\alpha_V = e^{-\frac{1}{\tau_V}\Delta t}$, and $\alpha_{I_{ext}} = e^{-\frac{1}{\tau_{decay}}\Delta t}$.

Similar to the rate-based model, we decomposed the recurrent weight matrix $W^{rec}$ as the product between a trainable non-negative matrix $W^{rec,+}$ and a fixed diagonal matrix $D$ (Song et al., 2016):

$$W^{rec} = W^{rec,+} D \tag{27}$$

For the forward spike function, we use the Heaviside function to generate the spike:

$$\text{spike}(x) = H(V[t] - V_{th}) = H(x), \tag{28}$$

where $x$ is used to represent $V[t] - V_{th}$.

To make the non-differentiable spiking activation compatible with the back-propagation algorithm, we considered a surrogate gradient:

$$\text{spike}'(x) = \text{ReLU}(\alpha * (\text{width} - |x|)) \tag{29}$$

where $\text{width} = 1.0$, and $\alpha = 0.3$. $\alpha$ is the parameter that controls the altitude of the gradient, and $\text{width}$ is the parameter that controls the width of the gradient. The shape of this function is shown in Figure S11.

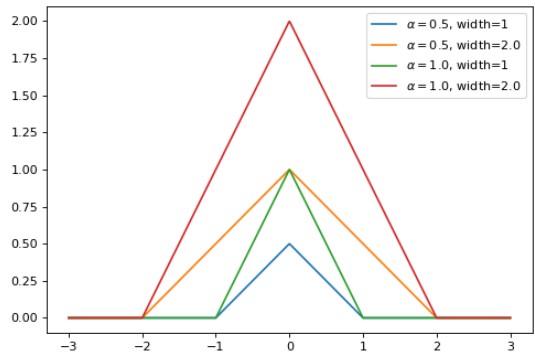

Figure S11: The shape of the ReLU gradient function depends on its parameters.

**The readout layer.** The recurrent neurons project linearly to the output neurons. For the rate-based model, the readout layer is given by:

$$\mathbf{y}[t] = W^{\text{out}}\mathbf{r}[t] + b^{\text{out}}, \qquad (30)$$

where $W^{\text{out}}$ are the synaptic weights between the recurrent and output neurons, and $b^{\text{out}}$ the bias.

For the spiking network, the output neuron is the leaky neuron, whose dynamics is given by:

$$\tau_{\text{out}}\frac{d\mathbf{y}}{dt} = -\mathbf{y} + W^{\text{out}}\mathbf{z} + b^{\text{out}}, \qquad (31)$$

where $\tau_{\text{out}}$ is the time constant of the output neuron, $W^{\text{out}}$ the synaptic weights between the recurrent and output neurons, and $b^{\text{out}}$ the bias. In the discrete description, the output dynamics is written as:

$$\mathbf{y}[t + \Delta t] = \alpha_{\text{out}}\mathbf{y}[t] + (W^{\text{out}}\mathbf{z}[t] + b^{\text{out}})\Delta t, \qquad (32)$$

where $\alpha_{\text{out}} = e^{-\frac{1}{\tau_{\text{out}}}\Delta t}$.

**Weight initialization.** Initial input, recurrent, and readout weights were drawn from a Gaussian distribution $W_{ji} \sim \sqrt{\frac{s}{n_{\text{in}}}}\mathcal{N}(0, 1)$, where $n_{\text{in}}$ is the number of afferent neurons, $\mathcal{N}(0, 1)$ is the zero-mean unit-variance Gaussian distribution, and $s$ is the weight scale. For inhibitory neurons, $s = 0.2$; for excitatory neurons, $s = 0.05$.

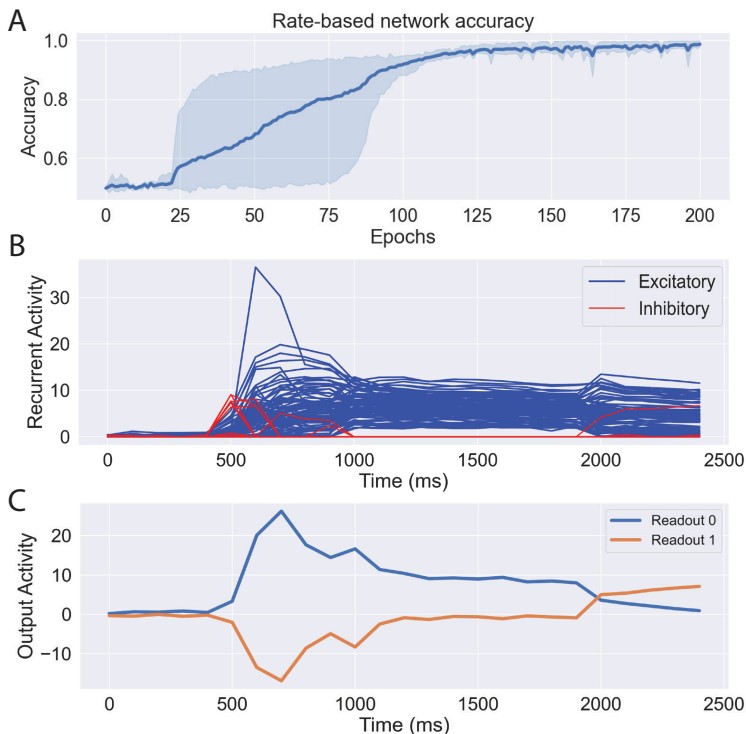

Figure S12: The rate-based recurrent model for implementing a DMS working memory task. (A) The training accuracy of the rate-based recurrent neural network for performing a DSM working memory task. (B) The recurrent activity during the network receiving a *non-match* case stimulus. (C) The output activity of the network when receiving a *non-match* case stimulus.

**Training methods.** The training was performed using the BPTT algorithm. The integration time step $\Delta t$ is 1 ms for the spiking neural network, while $\Delta t$ is 100 ms for the rate-based model. The Adam optimizer (Kingma & Ba, 2014) was used for computing the gradient-based optimization. The goal of the training was to minimize the cross-entropy between the output activity and the target output during the test period. For the spiking network, we add an additional voltage regularization loss that

penalizes voltages that fall significantly outside the support of the surrogate gradient function (Plank et al., 2021):

$$L_{\text{voltage}}[t] = \lambda_v \sum_i \sum_t (\text{ReLU}(V^i[t] - 0.4))^2 + (\text{ReLU}(-V^i[t] - 2.0))^2, \qquad (33)$$

where $\lambda_v$ is the strength of the voltage regularization.

**The performance of the rate-based model.** In contrast to the spiking-based model, the rate model employed here exhibits a slow convergence of training (see Figure S12A). It requires dozens of epochs of training to achieve a high accuracy. Figure S12 B and C also show the recurrent and output activities after the rate-based network receives a *non-match* case stimulus. Same as in the spiking network, the neural activity of the rate model during the delay period (1000 - 2000 ms) maintains a high firing rate.

## J   EVALUATION OF BRAINPY ON TRAINING MACHINE-LEARNING ORIENTED SPIKING NEURAL NETWORKS

To evaluate the performance of BrainPy on BIC models, we conduct a comparative analysis with several popular SNN training frameworks, including snnTorch (Eshraghian et al., 2021), Norse (Pehle & Pedersen, 2021), and SpikingJelly (Fang et al., 2020). Note here our comparisons are only carried out using densely connected spiking neural networks, which are predominantly utilized in existing BIC models.

We first use a simple three-layer SNN model (Neftci et al., 2019), where the input layer delivers synaptic currents to the hidden LIF layer with exponential dynamics, then the output layer readouts the hidden dynamics with an exponential synapse. The connections between each layer are dense layers. The training was performed using the BPTT algorithm on the Fashion-MNIST dataset (Xiao et al., 2017). The comparison result of the training speed per epoch is presented in Table S2. We find that BrainPy exhibits superior performances on both CPU and GPU devices. Furthermore, to test the performance of BrainPy on large-scale SNN networks, we re-implement a VGG SNN network (Xiao et al., 2022) which was originally coded in a PyTorch+SpikingJelly environment. The evaluation result is shown in Table S2. Under the same training setting, BrainPy achieves much better running performance. It reduces almost half of the time for training on GPUs.

Table S2: Comparison of average training speed per epoch for two SNN models using snnTorch (Eshraghian et al., 2021), SpikingJelly (Fang et al., 2020), Norse (Pehle & Pedersen, 2021), and BrainPy.

| Model | Device | snnTorch | SpikingJelly | Norse | **BrainPy** |
|---|---|---|---|---|---|
| Simple SNN (Neftci et al., 2019) | CPU (Intel) | $44.1 \pm 0.3$ s | $49.9 \pm 1.0$ s | $52.2 \pm 0.2$ s | $\mathbf{28.6 \pm 1.0}$ s |
| | GPU (RTX A6000) | $46.6 \pm 1.0$ s | $53.1 \pm 0.9$ s | $49.6 \pm 0.3$ s | $\mathbf{17.1 \pm 0.6}$ s |
| VGG SNN (Xiao et al., 2022) | GPU (RTX A6000) | - | $104.0 \pm 1.0$ s | - | $\mathbf{50.0 \pm 1.0}$ s |

## K   EVALUATION OF COMPILATION TIME BETWEEN BRAINPY AND BRIAN2

Simulating biological spiking neural networks often involves a time-consuming compilation process. To evaluate the compilation cost of BrainPy, we conducted a systematic comparison with Brian2 (Stimberg et al., 2019).

To assess the compilation time, we utilized two different network models: a simple EI balance network (Vogels & Abbott, 2005) and a more complex multi-area network model consisting of 30 interconnected brain areas (Joglekar et al., 2017).

During the initial run, both BrainPy and Brian2 compile the model. Consequently, we measured the compilation time by simulating the respective network for a single time step ($T = \Delta t$, where $T$ represents the simulation duration, and $\Delta t$ is the simulation time step).

For Brian2 on CPU, we measured the compilation time using the `cython` backend. To measure GPU compilation time, we utilized Brain2CUDA (Alevi et al., 2022).

In the case of the EI balance network, we varied the network size by increasing the number of neurons. We measured the compilation time across different network sizes and computing platforms. The comparison results are presented in Figure S13A. BrainPy demonstrates compilation speeds more than ten times faster than Brian2, regardless of the CPU or GPU device used. Brian2CUDA shows a gradual increase in compilation cost on GPU devices, which is consistent with the findings in their original paper. In BrainPy, CPU and GPU exhibit comparable compilation speeds.

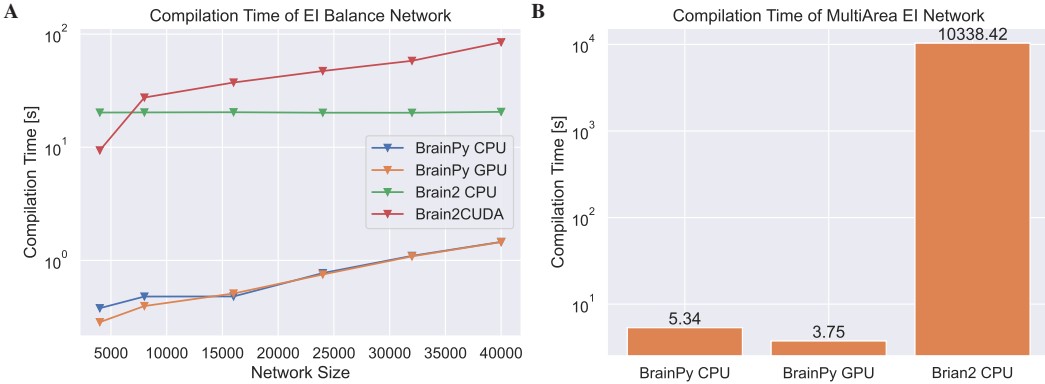

Figure S13: The compilation time comparison between BrainPy and Brian2. (A) The compilation time measured on an EI balance network model (Vogels & Abbott, 2005). (B) The compilation time measured on multi-area large-scale network model (Joglekar et al., 2017).

With the multi-area network model, BrainPy exhibits a significant speedup compared to Brian2 (Figure S13B). In this network, we employ `jax.vmap` to merge multiple projections from one brain area, resulting in a much simpler computational graph. In contrast, although the simulation of the network took only around one minute, the compilation process in Brian2 took over an hour. Note that in this model, the Brain2CUDA failed so we ignore the comparison with Brian2's GPU backend. This result unequivocally demonstrates the superior advantage of BrainPy, which leverages modern just-in-time compilation technology, over traditional brain simulators.

The compilation time comparisons between BrainPy and Brian2 on both the EI balance network and the more complex multi-area network demonstrate clear advantages of BrainPy's JIT compilation approach. BrainPy exhibited over 10x faster compilation times than Brian2 on both CPU and GPU devices, with the advantage being more pronounced on larger networks.

The gradual increase in Brian2's compilation time on GPU devices has been reported before (Alevi et al., 2022) and is likely due to inefficiencies in translating imperative Python code to GPU kernels. In contrast, BrainPy's compilation speed was consistent across CPU and GPU. This shows the power of just-in-time compilation and using computational graph optimizations like `jax.vmap` to simplify projections between areas.

The extremely long (>1 hour) compilation time for Brian2 on the multi-area network compared to the short simulation runtime (~1 minute) highlights a key bottleneck. As researchers continue building larger and more complex brain network models, short compilation times are essential to enable rapid testing and iteration. The over three orders of magnitude speedup shown by BrainPy is thus hugely impactful.

Overall, BrainPy's modern compilation approach enables faster development cycles and more complex networks than previously feasible. As brain models continue increasing in scale and detail, optimized just-in-time compilation will become more and more critical. These benchmark results validate BrainPy's advantages in compilation efficiency today and its ability to scale up to the brain models of tomorrow.

## L    SUPPORT FOR THE DISTRIBUTED COMPUTATION IN BRAINPY

BrainPy leverages JAX's parallel computation capabilities to enable distributed simulation and training of biological spiking neural network models. Specifically, it utilizes JAX's `pjit` interface which parallelizes execution across devices using XLA's Global Synchronous Parallelism for Multi-Device (GSPMD) protocol. This allows seamless model parallelism in BrainPy, wherein a single biological neural network can be partitioned across multiple devices with synchronized updates.

To showcase BrainPy's distributed computing capabilities, we provide an example using a decision-making neural network model from Wang (2002). This model has multiple interconnected brain areas that can be parallelized across devices. By using `pjit`, we are able to accelerate the simulation of this complex model by distributing it over multiple GPUs or TPUs. The difference under such parallelization context is that users should specify one `sharding` parameter when defining the model (for details please see the code in Data availability). Each device simulates a part of the network in sync with the other devices. This enables faster experimentation with larger models, as shown in Figure S14.

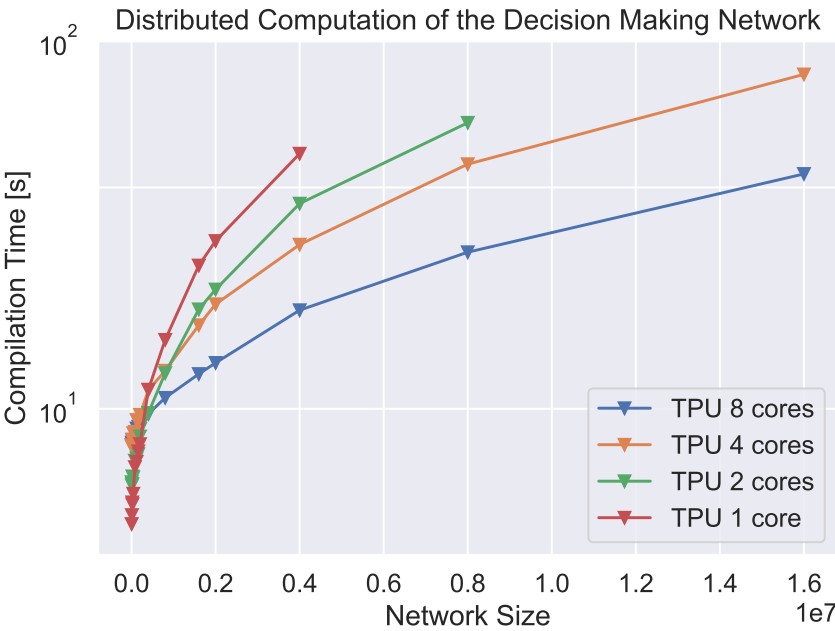

Figure S14: Distributed computation of a decision-making network (Wang, 2002) using BrainPy on TPUv3 devices. For TPU experiments with 1 and 2 cores, data are truncated due to the out-of-memory error.

It's worth noting that this distribution strategy has been successfully used in training large-scale DNN models, but it can incur significant communication overhead for spiking neural networks. This is because spike-based models tend to have sparse activations - with only a small subset of neurons spiking at each timestep. However, GSPMD synchronization requires gathering and broadcasting the full activation state across devices at every step, even though most values are zero.

This overhead is negligible for dense DNN activations but grows substantially with increasing sparsity levels. For example, in extreme cases where less than 1% of neurons in a spiking model spike per timestep, over 99% of the synchronization communication would transfer redundant zero values. This redundant communication consumes network bandwidth and limits scalability.

To address this inefficiency, we are exploring more optimized synchronization schemes tailored to spiking neural networks. These could leverage compression or gather only non-zero spikes to reduce communication costs. Alternately, asynchronous update approaches could be employed that relax strict lockstep synchronization between devices. Such spike-specific optimizations can unlock greater scalability for large-scale spiking models in BrainPy while retaining its flexible device distribution capabilities.

## M    PACKAGE STRUCTURE

BrainPy is a comprehensive Python library for modeling and simulating brain dynamics. It provides a flexible and extensible framework for building, simulating, training, and analyzing biological neural networks and brain-inspired algorithms. BrainPy is built on top of efficient Just-In-Time (JIT) compilers, enabling high-performance computations for various brain dynamics models.

The BrainPy package is organized into several core modules that encapsulate different aspects of brain dynamics programming:

- `brainpy.math`: This module provides a collection of mathematical functions and utilities for use in brain dynamics modeling. It includes functions for numerical operations, vector and matrix operations, random number generation, and dedicated event-driven, sparse, and JIT connectivity operators.

- `brainpy.integrators`: This module provides a variety of numerical integration methods for solving diverse differential equations (including ordinary differential equations, stochastic differential equations, fractional differential equations, and delay differential equations) that arise in brain dynamics models. These solvers are based on well-established numerical techniques, such as Euler's method, Runge-Kutta methods, and adaptive solvers. The `brainpy.integrators` module is an essential component of the BrainPy library, providing a powerful and versatile set of tools for integrating differential equations used in brain dynamics models.

- `brainpy.dyn`: This module provides fundamental building blocks for biological neural networks, including ion channels, neurons, synapses, projections, plasticity models, populations, and networks.

- `brainpy.dnn`: This module provides a high-level API for constructing deep neural networks. It is designed to be simple and easy to use, while still providing the flexibility and power needed to build complex models. The module includes a variety of pre-built layers, including convolutional layers, pooling layers, and fully connected layers. The DNN models can also be used as a component in `brainpy.dyn` models.

- `brainpy.analysis`: The module provides a collection of tools for analyzing the dynamics of neural networks, including phase plane analysis and bifurcation analysis for low-dimensional dynamical systems, and linearization analysis for high-dimensional ones.

- `brainpy.train`: This module provides algorithms for training biological neural networks, including supervised learning, online learning, and offline learning.

- `brainpylib`: This module provides customized operators and utilities for optimizing brain dynamics computations.

In addition to these core modules, BrainPy also includes a toolbox that extends its core functionality:

- `brainpy.measure`: The module provides a collection of functions for measuring and evaluating the properties of neural network models, such as firing rates, synchronization, and functional connectivity.

- `brainpy.inputs`: The module provides a collection of functions for defining and handling input data for models in BrainPy.

- `brainpy.connect`: The module provides a mechanism for connecting neurons and synapses in BrainPy. It allows users to specify the connectivity pattern between different populations of neurons and to define the properties of synaptic connections, such as weights.

- `brainpy.initialize`: The module provides a collection of functions for initializing the parameters of models in BrainPy. It includes various initialization schemes, such as random initialization, Xavier initialization, and He initialization. It also provides functions for initializing other network components, such as biases and noise sources.

For more details about BrainPy's package structure, please see Figure S15.

```
BrainPy
├─ analysis              │ Brain dynamics analysis tools for differential equations.
│  ├─ highdim            │ linearization analysis and fixed/slow point finding.
│  └─ lowdim             │ Phase plane analysis, bifurcation analysis, and fast-slow bifurcation analysis.
├─ checkpoints           │ Serialization, save and load the variables of a model.
├─ connect               │ Construct synaptic connection, inclluding built-in and self-customized connectors
│  ├─ custom_conn.py     │ Construct self-customized connections.
│  ├─ random_conn.py     │ Construct random connections.
│  └─ regular_conn.py    │ Construct regular pattern connections.
├─ dnn                   │ Contain layers in deep learning and interoperation with Flax.
├─ dyn                   │ Contain classes and functions describe the dynamics of neural systems.
│  ├─ ions               │ Ions include sodium (Na+), potassium (K+), calcium (Ca2+), and chloride (Cl-).
│  ├─ channels           │ Ions channels dynamics inluding many implementations.
│  ├─ neurons            │ Neural dynamics including simplified and biophysical neurons.
│  ├─ synapses           │ Synaptic Dynamics including phenomenological, biological synapses, gap junctions.
│  ├─ projections        │ Synaptic projections including AlignPre, AlignPost and STDP.
│  ├─ rates              │ Firing rate models including population rate models, RNN, and reservoir computing
│  ├─ outs               │ Synaptic outputs including CUBA, COBA, and MgBlock.
│  └─ others             │ Common dynamics models including leaky integrator, input/output groups.
├─ dynold                │ Compatible with previous brain dynamics models interface.
├─ encoding              │ Encoding rate values as spike trains.
├─ initialize            │ This module provides methods to initialize weights.
├─ inputs                │ Calculate an input current with different formats.
├─ integrators           │ Numerical integrators including ODEs, SDEs, FDEs and DDEs.
├─ losses                │ Common losses used in machine learning tasks.
├─ math                  │ Provides the mathematical foundation for brain dynamics programming.
│  ├─ delay              │ Delay variables.
│  ├─ event              │ Event-driven operators.
│  ├─ jitconn            │ JIT connections operators.
│  ├─ sparse             │ Sparse operators.
│  ├─ object_transform   │ Object-oriented Transformations.
│  ├─ op_register        │ Operators registrations.
│  ├─ surrogate          │ Surrogate gradients for gradient-based learning.
│  └─ sharding           │ Parallelization Support for model sharding.
├─ measure               │ Measurement for calculating correlations and firing rates.
├─ optimizers            │ Gradient-based optimizers used in deep learning.
├─ running               │ Parallel simulation techniques with multi-processes and multi-devices.
├─ tests                 │ Assemble tests.
├─ tools                 │ Addtitional tools used for brain dynamics programming.
├─ train                 │ Various running and training algorithms for various neural networks.
│  ├─ back_propagation.py│ Back-propagation learning method
│  ├─ offline.py         │ Offline training methods, like ridge regression, linear regression, etc.
│  └─ online.py          │ Online training methods, like RLS, LMS, etc.
├─ mixin                 │ Mixin objects provide behaviour used for brain dynamics programming.
└─ visualization         │ Visualization toolkit.
```

Figure S15: The structure and components of BrainPy

## N  IMPLEMENTATION OF EI BALANCE NETWORK USING THE JIT CONNECTIVITY OPERATORS

For the details of the EI balance network, please refer to (Vogels & Abbott, 2005). To implement this balanced network in BrainPy, we leverage the just-in-time (JIT) connectivity operators (see Section 4.2) to dynamically create connections between excitatory and inhibitory neuron populations during the simulation. We first initialize the neuron population specifying the neuron models and population sizes (see Line 12 in Listing S6). Then we use JIT connectivity to randomly connect the two layers with a given sparsity and specified weight for excitatory-to-excitatory, inhibitory-to-inhibitory, and inhibitory-to-excitatory projections (see Lines 17, 18, 24, and 25 in Listing S6). Note that `brainpy.dnn.EventJitFPHomoLinear` has implemented an event-driven matrix multiplication JIT operator with homogeneous synaptic weight. Specifically, this layer is based on the operator `brainpy.math.jitconn.event_mv_prob_homo`.

Key parameters to tune include the maximum connection synapse per neuron and connection weight strengths. Monitoring the relative excitatory and inhibitory firing rates over time allows assessment of EI balance and guides further parameter adjustments towards stabilizing the network dynamics. The JIT connectivity functionality has the same workflow for simulation in BrainPy.

```
1 import brainpy as bp
2 import brainpy.math as bm
3
4 class EINet(bp.DynSysGroup):
5   def __init__(self, scale=1.):
```

```
6      super().__init__()
7      self.num = int(4000 * scale)
8      self.num_exc = int(3200 * scale)
9      self.num_inh = int(800 * scale)
10     p = 80 / self.num
11
12     self.N = bp.dyn.LifRef(self.num, V_rest=-60., V_th=-50.,
13                            V_reset=-60., tau=20., tau_ref=5.,
14                            V_initializer=bp.init.Normal(-55., 2.))
15     self.delay = bp.VarDelay(self.N.spike, entries={'I': None})
16     self.E = bp.dyn.ProjAlignPostMg1(
17       comm=bp.dnn.EventJitFPHomoLinear(self.num_exc, self.num,
18                                        prob=p, weight=0.6),
19       syn=bp.dyn.Expon.desc(size=self.num, tau=5.),
20       out=bp.dyn.COBA.desc(E=0.),
21       post=self.N
22     )
23     self.I = bp.dyn.ProjAlignPostMg1(
24       comm=bp.dnn.EventJitFPHomoLinear(self.num_inh, self.num,
25                                        prob=p, weight=6.7),
26       syn=bp.dyn.Expon.desc(size=self.num, tau=10.),
27       out=bp.dyn.COBA.desc(E=-80.),
28       post=self.N
29     )
30
31   def update(self, inp):
32     spk = self.delay.at('I')
33     self.E(spk[:self.num_exc])
34     self.I(spk[self.num_exc:])
35     self.delay(self.N(inp))
36     return self.N.spike.value
37
38
39 model = EINet(scale=10.)
40 indices = bm.arange(1000)   # 100 ms
41 spks = bm.for_loop(lambda i: model.step_run(i, 20.), indices)
42 bp.visualize.raster_plot(indices, spks, show=True)
```

Listing S6: Implementation the EI balance network model using the JIT connectivity operators.

## O    DATA AVAILABILITY

BrainPy is a pip installable Python package and available at the following GitHub repository: `https://github.com/brainpy/BrainPy`, with documentation at `https://brainpy.readthedocs.io/`. All the code to reproduce the result presented in the paper can be found in the following GitHub repository: `https://github.com/brainpy/brainpy-iclr24-reproducibility`.

The MNIST dataset can be found in `http://yann.lecun.com/exdb/mnist/` and it is also conveniently accessible in Python via the `brainpy-datasets` package. The processed KTH data can be available in (Antonik et al., 2019). The multi-area connectivity data can be found in (Markov et al., 2014) and `https://core-nets.org/`.

## P    SUPPLEMENTARY FIGURES

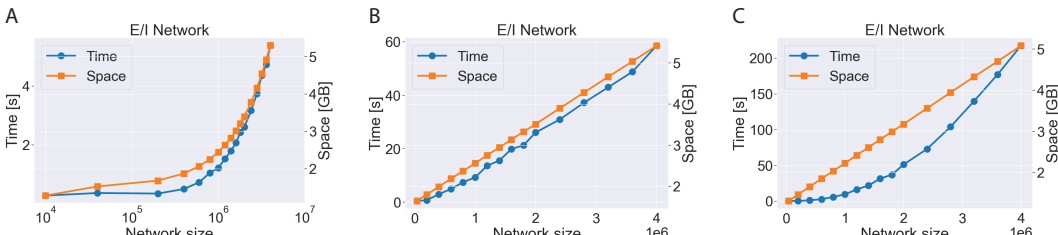

Figure S16: Additional experiments of E/I balanced network. (A) Depicting the data from Figure 5C on a log-scale x-axis. (B) E/I balanced network simulation with a thousand synapses per neuron. Under this condition, the network still shows the linear scaling property. (C) E/I balanced network simulation where a fixed connection probability (p=0.001) is used and the weight is rescaled.

