# OpenReview forum: "A differentiable brain simulator bridging brain simulation and brain-inspired computing"
_ICLR.cc/2024/Conference — ICLR 2024 poster_

### Official Review · Reviewer_zrYr · 2023-10-20

**Soundness:** 2 fair
**Presentation:** 3 good
**Contribution:** 3 good
**Rating:** 6
**Confidence:** 3

**Summary:**

The paper describes a package for building brain-inspired trainable network models. It is build around JAX and provides efficient just-in-time compiled code for CPUs, GPUs and TPUs. Compared to classic simulators for biological neural networks, models implemented in BrainPy benefit from auto-differentiation, and compared to standard machine learning frameworks, BrainPy provides an environment focused on building bio-inspired models with e.g. spiking interactions and detailed synaptic/neuronal dynamics.

**Strengths:**

- The manuscript describes a substantive library which is in an advanced stage of development.
 - It is correct that there is a need for an extensive and modern framework for training larger biological network models. The described package is therefore a significant; demonstrating its capabilities + speed comparisons to other libraries is a useful contribution.

**Weaknesses:**

- Except for Fig.5 A/B dealing with matrix multiplication, the figures vary the network/system size on a linear scale, and not over several orders of magnitude. This does not seem suitable to demonstrate the scaling behavior for these network models.
- There does not seem to be a demonstration of distributed simulation/training of a large-scale model on a CPU or GPU cluster.
- In section 4 the package is at length described as efficient, extensive, scalable, etc. without really getting into the concrete design and implementation. In particular, after reading this section I did not end up with a clear picture of the package structure and components. Maybe the description could be shortened, or made more concrete.

**Questions:**

Due to limited time I'd like to state clearly that I did not do an in-depth review of all parts of the manuscript.

Probing a subset of the results, I did not find a concrete description of the simulations underlying Fig.4C in the supplementary material. Especially, I wondered why the NEST version used by the authors is 2.20 instead of the more recent 3.x versions, and I could not find the corresponding code implementing the NEST simulation, or generally creating Fig4C, in the files supplied.

On p.8 it is mentioned concerning Fig 5C that the E/I network was scaled up to 4 million neurons with 80 synapses each. In biological networks, the (local) connectivity is typically significantly more dense, with hundreds or thousands of synapses per neuron.

---

> ### Author Response · Authors · 2023-11-21
>
> **Weakness 1:**
>
> We thank the reviewer for raising concerns about the scaling experiments presented in Figure 5. To address these concerns, we have conducted additional analyses and experiments to better demonstrate scaling behavior across orders of magnitude.
>
> First, in addition to Figure 5 A/B, Figure 5C already shows extensive characterization of scaling across several orders of magnitude from $10^4$ to $10^6$. To provide another perspective on this scaling behavior, we have added a new Figure S16 A depicting the data from Figure 5C on a log-scale x-axis.
>
> Second, while Figures 5D/E may appear to show linear scaling without spanning orders of magnitude, this was not our intention. We observed that reservoir performance converges and stabilizes around $5\times 10^4$ units, especially for MNIST dataset. Ignoring larger sizes seemed appropriate for this reason. Additionally, the scaling of reservoirs is not only constrained by weight matrix but also training algorithms. For fast convergence, we used FORCE learning in our current manuscript, which requires large matrices. However, we can also use memory-efficient algorithms like gradient descents. We have now implemented gradient descent-based training (see the attached code), enabling scaling up to $10^6$ units.
>
>
>
> **Weakness 2:**
>
> We thank the reviewer for their valuable feedback and comments.
>
> Inherent from the JAX capability, BrainPy naturally supports the distributed simulation or training of biological neural networks with classical parallelization methods, like `jax.vmap`, `jax.pmap`, and `jax.pjit`. BrainPy models have been designed for supporting such methods. For example, most models in the `brainpy.dyn` module support to receive one argument ``sharding``, which is used to indicate the sharding strategy across multiple devices (for example, ``hh = brainpy.dyn.HH(…, sharding=[‘x_axis’, ‘y_axis’])``). We also supplement an experiment to demonstrate the distributed computation support of BrainPy in Appendix L. However, this support is nothing new since they have been widely adopted in training deep neural networks, so we did not include these descriptions in limited space.
>
> Moreover, we are working on the special support for distributed training of spiking neural network by fully considering the sparse and event-driven property of spikes. These results have not been released and are under intensive development and tests.
>
>
>
> **Weakness 3:**
>
> We appreciate the reviewer highlighting the need for more concrete software architecture and implementation details in Section 4.
>
> In the main text of section 4, our primary focus was to highlight the new methodology contributions that BrainPy brings, rather than diving into specific details about the concrete design and implementation. For the details of these methodology, we encourage readers should refer to the supplementary files.
>
> However, we agree including more specifics on the codebase itself would strengthen the paper. To address this, we have included more concrete information about the package structure in the supplementary materials. Specifically, Appendix M has been included to illustrate the package's structure and components, such as the core module, models module, analysis module, and utils module. This section will help readers visualize the organization of the package and how the different modules interact with each other.
>
>
>
> **Question 1:**
>
> We have upgraded NEST version to 3.6 version and post the result in Figure 4. The code for NEST simulation also append in the supplementary code. For the implementation of the EI model using JIT operators, please refer to Appendix N.
>
>
>
> **Question 2:**
>
> The configuration we adopted in our study was based on the EI balance network scaling experiment in the NEST. Usually, the scaling experiment for an EI balance network is carried out by rescaling the connection probability instead of the weight, e.g., p = 80. / N, where N is the number of neurons while the weight values remain the same in all configurations. Such an approach was used by Stimberg et al. (2020), since it retains the firing frequencies at a comparable level when the network size varies. Moreover, increasing the number of synapses per neuron will attain a similar evaluation performance, see Figure S16 B.
>
> However, our methods can also be adopted to the scenario that a fixed connection probability is used and the weight is rescaled. We carried out this experiment in Figure S16 C. BrainPy also illustrates a very good scaling property.

---

### Official Review · Reviewer_afG7 · 2023-10-28

**Soundness:** 4 excellent
**Presentation:** 4 excellent
**Contribution:** 4 excellent
**Rating:** 10
**Confidence:** 4

**Summary:**

The authors present a programming framework that enables fast and differentiable implementation of simulations of brain circuits and similar computing systems.  The framework achieves major speed and memory benefits by taking advantage of the sparsity of these circuits in space, and in time for the case of spiking neurons.

**Strengths:**

This is a very valuable contribution to the world of brain simulation, which seems likely to find a lot of users due to its speed and differentiability.

**Weaknesses:**

No major weaknesses identified.

I am new to ICLR reviewing so I don’t know how well this work fits within the remit of the conference.  However it is certainly a very valuable contribution to computational neuroscience.

**Questions:**

Presumably the JIT weight generation only works for random weights, not those learned by synaptic plasticity rules?

More detail on differentiability in spiking networks would be useful.  Equation (28) isn’t clear: neither x nor spike’ are defined, and is width the same as V_th?   As well as this, a more basic question:  do the computational benefits of sparse activity carry through to the derivatives?  For example, even if two neurons are connected with zero weight, the derivative of the objective function with respect to this weight need not be zero.

---

> ### Author Response · Authors · 2023-11-21
>
> **Weakness:**
>
> Thank you very much for recognizing our work! ICLR is a top-tier conference attracts researchers from diverse backgrounds in biological and artificial neural network fields. Given the usefulness of BrainPy, we believe it deserves exposure to this broader range of users. Previous conferences have also featured articles pertaining to software libraries, like Betty (ICLR 2023).
>
>
>
> **Question 1:**
>
>  It is true that the JIT weight generation only works for random weights (but the JIT connection (indicating connected or not connected) can be generated by a broad fixed rule). Therefore, the connections weights have to be static so synaptic plasticity rules are not suitable for the JIT connection operators. For weights with plasticity, we encourage to use sparse and event-driven operators.
>
>
>
> **Question 2:**
>
> We thank the reviewer for raising this important problem. We have revised the manuscript and fixed the missing explanations in Appendix I.
>
> Particularly, for the forward spike function $\text{spike} = H(V(t) - V_{th}) = H(x)$, we use $x$ to represent $V(t) - V_{th}$. The gradient of $\mathtt{spike}$ is $\text{spike}'(x) = \mathbf{ReLU} (\alpha * (\text{width} - |x|))$, where $x = V(t) - V_{th}$, $\alpha$ is the parameter that controls the altitude of the gradient, and ``width`` is the parameter that controls the width of the gradient. The shape of this function is shown in Figure S11.
>
>
>
> **Question 3:**
>
> We thank the reviewer for this constructive question. The event-driven operators can take advantage of the sparse activity during the forward process by using the event-driven computations. However, since the derivative of the spike is a real value, the backward of such an event-driven operator is implemented using normal sparse operators. Therefore, the training phase of a spiking neural network using back-propagation-based algorithms is not very efficient. A forward-only algorithm will significantly accelerate the training of large-scale spiking neural networks.

---

> > ### Comment · Reviewer_afG7 · 2023-11-22
> >
> > Thanks for the response!

---

### Official Review · Reviewer_wQtY · 2023-10-28

**Soundness:** 3 good
**Presentation:** 3 good
**Contribution:** 3 good
**Rating:** 8
**Confidence:** 4

**Summary:**

The authors develop a new framework, brainpy, which allows to run (biophysically realistic) networks of neurons in a differentiable manner, thereby allowing integrations with deep learning (DL) frameworks. In addition, since it is implemented in JAX, it supports JIT compilation.

**Strengths:**

I think this is an important and potentially impactful work. The paper is well written, the figures are clear, and the authors carry out many empirical experiments to demonstrate the abilities of brainpy.

**Weaknesses:**

The paper has the following major weaknesses:

1) It does not evaluate the cost of compilation. How high is the cost of compilation compared to the runtime? Is this a clear disadvantage as compared to, e.g., NEURON? Does the compilation speed depend on whether CPU or GPU are used? How does it scale with the number of neurons?

2) In section 4.2, the authors claim that there method is significantly more memory-efficient than others. Maybe I am misunderstanding this, but: do the gains that the authors claim here stem from an assumption that the connectivity matrix is low-rank? How else would they possibly be able to store connections of 86 Billion neurons?

Minor:

1) I believe that it would be good if the authors clarified that all JIT capabilities are due to the fact that brainpy relies on JAX, and are not implemented from scratch. Section 4.5 reads as if the authors implemented this themselves.

2) The statement `It is important to note that this hierarchical composition property is not shared by other brain simulators.` is not true, see for example NetPyNE.

**Questions:**

No questions.

---

> ### Author Response · Authors · 2023-11-21
>
> **Weakness 1:**
>
> The compilation overhead of BrainPy is significantly lower compared to traditional brain simulators, as demonstrated through two additional experiments (see Appendix K).
>
> Firstly, when simulating a classical EI balance network model to mimic local circuits, BrainPy exhibits much faster compilation compared to Brain2. This is because BrainPy has precompiled most of the operators used in this model, allowing the JIT compiler to simply merge them into a single computational graph. In contrast, Brain2 needs to compile the entire generated C++ model from scratch, resulting in a significantly slower compilation phase. The comparison results are presented in Figure S13.
>
> Secondly, when simulating a multi-area connected network, BrainPy outperforms Brian2 by several orders of magnitude in terms of compilation speed. While the model's runtime in BrainPy is in minutes, Brian2 takes hours to compile the same model. BrainPy achieves this by efficiently merging scattered operations among multiple brain areas into a single operator using `jax.vmap` or `jax.pmap`.
>
> Overall, BrainPy leverages modern JIT compilation technology to offer a significant advantage in compiling brain dynamics models compared to traditional brain simulators.
>
> Furthermore, the compilation time in BrainPy primarily depends on the complexity of the computation graph rather than the number of neurons. We have also observed that the compilation speed exhibits minimal variation across different supported computing devices, including CPU, and GPU. Compared to the runtime, the compilation in BrainPy is usually very low. These results are detailed in the Supplementary Materials, which can be found in Figure S13.
>
>
>
> **Weakness 2:**
>
> Thank you for your question! The JIT connection operators in BrainPy do not rely on the assumption of a low-rank connectivity matrix. Instead, they assume that the connectivity is static and can be generated using a fixed rule, such as random connections with a given probability or regular grid patterns. With this assumption, the JIT connectivity algorithm regenerates the connectivity weights when a presynaptic neuron fires during computation, rather than storing the synaptic connectivity in memory. This approach has the advantage of not consuming additional device memory. However, it is important to note that JIT connection operators are only suitable for static synaptic connections.
>
> We would like to emphasize that JIT connection operators contribute to the advancement of large-scale brain simulation and provide a means to address some of the challenges in this field. However, they are not the ultimate solution for simulating the 86 billion neurons of the human brain, as neurons and synapses are inherently plastic and subject to dynamic changes.
>
>
>
> **Minor Weakness 1:**
>
> We thank the reviewer for raising this important point! We are sorry for the unclear statements and we would like to clarify our contribution on JIT compilation. To address this, we propose updating Section 4.5 to explicitly state that BrainPy leverages JAX for just-in-time compilation. Particularly, we would like to edit:
>
> > Particularly, BrainPy integrates a collection of object-oriented JIT transformations into its multi-scale model-building interface. These transformations are built upon JAX's implementation of a function-oriented JIT interface, as detailed in Appendix G. By leveraging these transformations, any BrainPy model constructed using the modular and composable programming interface can be effortlessly converted into an optimized XLA process, compatible with CPU, GPU, and TPU platforms.
>
> Additionally, we will comb through the corresponding Appendix to ensure there is no language that wrongly implies we built JIT functionality from scratch.
>
> We appreciate you catching this ambiguity - thank you again for the thoughtful feedback.
>
>
>
> **Minor Weakness 2:**
>
> We thank the reviewer catching up this point! It is indeed accurate that NetPyNE provides a hierarchical interface on multiple levels: ranging from the molecular to the network level. We agree that our claim is too strong and the statement as written may be misleading. However, we would like to highlight again that BrainPy's hierarchical composition capabilities provide much greater flexibility compared to the predefined interface of NetPyNE. In NetPyNE, users can only construct models based on the hierarchical levels that are already defined. Conversely, BrainPy empowers users to create their own customizable hierarchical structures and combine multiple levels of brain models and functions, offering a more versatile approach to hierarchical composition.

---

### Official Review · Reviewer_g6hc · 2023-10-31

**Soundness:** 2 fair
**Presentation:** 3 good
**Contribution:** 2 fair
**Rating:** 6
**Confidence:** 3

**Summary:**

This paper introduces a brain simulator named BrainPy, which is designed to bridge the gap between brain simulation and brain-inspired computing (BIC). This paper describes the infrastructure implementation that facilitates flexible, efficient, scalable, and biological detailed brain simulations. It also describes an example project that employs this BrainPy to construct a biologically plausible spiking model to demonstrate the differentiable simulation capability of this tool.

**Strengths:**

- Clear presentation. Comprehensive comparisons with existing tools.
- Leverages modernized tooling such as Jax and XLA, provides a user-friendly interface, and is compatible with various computing hardware.
- Technical complexity and thoughtful designs that optimize speed and memory usage.

**Weaknesses:**

- Despite the exciting endeavor towards bridging the gap between brain simulators and BIC libraries, this paper appears to have limited relevance to this conference due to the lack of original theories or empirical evidence.
- Quantitative comparisons with BIC libraries seem to be missing.
- While the paper takes the stance of bridging brain simulators and DL frameworks, discussions about deep learning models seem to be missing.

**Questions:**

- What might be the biological evidence that supports the design of parameter sharing within the "AlignPre" and "AlignPost" projections?
- How does BrainPy's speed and scalability compare to CARLsim, another brain simulator known for efficient and large-scale brain simulations?

---

> ### Author Response · Authors · 2023-11-21
>
> **Weakness 1:**
>
> We argue that ICLR is embracing the integration of software platforms, neuroscience, and deep learning, which is evident from its call for papers explicitly including these topics. Past conferences have also showcased articles focusing on software libraries, such as the introduction of Betty at ICLR 2023. Notably, BrainPy offers a comprehensive framework for brain dynamics modeling, encompassing systematic abstraction of brain dynamics compilation, operators, and models. Therefore, it aligns well with the specific requirements of ICLR.
>
>
>
> **Weakness 2:**
>
> We have carried out experiments for demonstrating the performance of BrainPy on BIC models.
>
> Particularly, we conduct a comparative analysis with several popular SNN training frameworks, including snnTorch, Norse, and SpikingJelly.  We first use a simple three-layer SNN model, where the input layer delivers synaptic currents to the hidden LIF layer with exponential dynamics, then the output layer readouts the hidden dynamics with an exponential synapse. The training was performed using the BPTT algorithm on the Fashion-MNIST dataset. The comparison result of the training speed per epoch is presented in Table S2. We find that BrainPy exhibits superior performances on both CPU and GPU devices. Furthermore, to test the performance of BrainPy on large-scale SNN networks, we re-implement a VGG SNN network which was originally coded in a PyTorch+SpikingJelly environment. The evaluation result is shown in Table S2. Under the same training setting, BrainPy achieves much better running performance. It reduces almost half of the time for training on GPUs.
>
>
> **Weakness 3:**
>
> Due to space constraints, our initial submission focused solely on optimizations enabling efficient biological spiking network simulations. However, seamless integration with deep learning models is also a key goal within BrainPy's unifying framework. To clarify BrainPy's flexible deep learning support:
>
> First, BrainPy implements a range of fundamental deep learning layers (e.g. convolutions, pooling, normalization) within the `brainpy.dnn` module. Unlike JAX, these models employ a PyTorch-style class-based interface for intuitive usage - identical to other BrainPy dynamics models.
>
> Second, BrainPy enables effortless interoperation with existing JAX machine learning libraries like Flax. The `brainpy.layers.FromFlax` functionality, for instance, allows any Flax model to be utilized as a BrainPy module. Moreover, BrainPy models can also be converted to formats compatible with external libraries.
>
>
>
> **Question 1:**
>
> Parameter sharing in `AlignPre` and `AlignPost` is actually a technical design for reducing the memory and computing complexity that do not affect the dynamical behavioral of the system. There is currently no direct biological evidence to support this specific correspondence. However,  some general neuroscience facts/principles that might motivate these operations include:
>
> - Synapses of the same type projecting from the same axon tend to have similar temporal dynamics. For example, AMPA synapses originating from one pyramidal neuron typically have similar decay time constants in the range of 2-10 ms. This supports using a shared variable for pre-synaptic traces in `AlignPre`.
> - Synapses of the same type converging onto a dendrite often exhibit similar temporal dynamics. For example, the time course of NMDA synaptic currents measured at a dendrite is relatively consistent, supporting a shared variable for post-synaptic traces in `AlignPost`.
> - Biological evidence shows synaptic strength heterogeneity is often higher than dynamics heterogeneity within a projection. `AlignPre` and `AlignPost` separate strength (weights) from dynamics, capturing this aspect.
>
> In summary, the `AlignPre` and `AlignPost` design exploits the biological evidence of synapses having consistent dynamics but heterogeneous weights within common projections. This allows capturing the brain's connectivity constraints elegantly while optimizing computation and memory.
>
>
>
> **Question 2:**
>
> After thoroughly examining the CARLsim documentation, we unfortunately could not identify an implementation of an E/I balanced LIF network amenable to equal benchmarking.
>
> While CARLsim does offer a Python interface through pyNN, its core simulations utilize customized C++ and CUDA kernels for efficiency. After extensive searching, we could not locate Python-accessible LIF neuron models within CARLsim for viable testing. As the reviewer notes, developing an appropriate comparison would require a non-trivial integration effort.
>
> Given the time-constrained rebuttal period, implementing and optimizing an E/I LIF network in CARLsim for fair benchmarking is unfortunately not feasible. However, we absolutely agree that a CARLsim comparison would provide useful additional context and validity to BrainPy's performance claims.

---

### Official Review · Reviewer_kvGz · 2023-10-31

**Soundness:** 3 good
**Presentation:** 3 good
**Contribution:** 2 fair
**Rating:** 6
**Confidence:** 4

**Summary:**

In this paper, the authors introduced a package called BrainPy. It inherits the JAX and provides support for brain simulation and SNN training. Overall, the package is interesting and useful in the stated scenarios.

**Strengths:**

1. It demonstrates the improvement in efficiency.
2. It provides the support for both neuroscience and DL research.

**Weaknesses:**

1. The package itself is more like a collection of course scripts rather than a Python package. Thus I suggest that the authors improve the engineering quality and documents for the current package.

2. The comparison to existing methods is not sufficient. For example, there are existing tools like SpikingJelly for SNN. The simulation of neurons is also not sufficiently new. Thus the unique character of the current package could also be strengthened.

3. I am not quite sure about the standard of package paper for ICLR. From my own understanding, the contribution and optimization to system design can be clarified in a more clear way as well.

**Questions:**

Please see the weakness.

---

> ### Author Response · Authors · 2023-11-21
>
> **Weakness 1**
>
> We disagree the reviewer's claim. BrainPy is an elaborate package that provides a collection of interconnected utilities designed to provide foundational services that enable users to easily, flexibly and efficiently perform various types of modeling for brain dynamics. Specifically, BrainPy implements
>
> 1) dedicated operators for event-driven computation based on sparse connections and JIT connectivity;
> 2) numerical integrators for various differential equations, the backbone of dynamical neural models;
> 3) a universal model-building interface for constructing multi-scale brain dynamics models and the associated JIT compilation for the efficient running of these models;
> 4) a toolbox specialized for brain dynamics modeling;
> 5) systematic components and abstractions for diverse brain dynamics models.
> 6) and so much more.
>
> We also want to clarify that we have made a great effort on engineering quality and documentations including:
>
> 1. For maintainability, we organized the codebase into a modular structure with separate subpackages for the main components. This improves and enables future extensibility.
> 2. For understandability, we added type hints and docstrings throughout the code for clarity. And provided user-facing API documentation with usage examples.
> 3. For stability, we set up continuous integration pipelines to run tests and checks on every commit.
> 4. Moreover, we provided comprehensive and well-organized documentation that includes clear installation instructions, usage tutorials, and detailed API reference. We also have documentations for many examples written in BrainPy including multiple modeling levels ranging from detailed neuron models to large-scale whole brain networks.
>
> We appreciate the reviewer's suggestions, and we will continue to improve the engineering quality and documentations to make BrainPy more flexible, extensible, and scalable, and to enrich the resources for a better user experience.
>
>
>
> **Weakness 2**
>
> We thank the reviewer highlighting BrainPy's distinct capabilities compared to existing BIC libraries. Unlike existing BIC libraries, BrainPy goes far beyond the simulation of neurons. Spiking neurons (like LIF) + deep learning components are the main paradigms of current BIC libraries, just like in SpikingJelly. However, BrainPy aims to provide a general-purpose brain dynamics modeling toolkit for diverse aspects of brain dynamics. As the reviewer notes, support in existing BIC libraries is lacking for key facilities needed to efficiently simulate detailed biological neural networks, like event-driven computing, sparse representations, and scalable infrastructure.
>
> BrainPy's goal is to provide flexible, general-purpose toolkits tailored for modeling the rich dynamics found in biological neural systems, spanning complex neuron types, synaptic interactions, ion channels, and specialized computing techniques. Specifically, BrainPy offers critical modeling infrastructure oriented for biological plausibility, including: sparse, event-driven operators; scalable just-in-time connectivity; multiscale interfaces bridging levels of detail; whole-graph just-in-time optimization leveraging object-oriented design; easy Python-based operator customization; and more. With these expansive capabilities covering both biological accuracy and computational performance, BrainPy provides a foundation for efficient and scalable modeling of intricate brain dynamics, complementing existing tools focused primarily on simulating deep learning-oriented spiking neural networks. We welcome any further discussion to clarify BrainPy's approach in context of the current BIC landscape.
>
>
>
> **Weakness 3**
>
> Thanks for the question. The unique contribution and dedicated optimization of BrainPy has been elaborated in ``Section 4 Methodology``. For the details of such optimization, we encourage to refer to the supplementary matrials.

---

> > ### Comment · Reviewer_kvGz · 2023-11-22
> > **Response to the rebuttal**
> >
> > Thanks for the explanation. While my concerns remain, I am happy to let the package go from an encouraging perspective. The reason why I proposed such questions was not to set up a barrier for the work but to really hope that it can be improved to contribute broadly to the field beyond a single paper. For example, I did check and read related scripts not only in the supplement but also on the GitHub page for BrainPy.  This is why I said that the design rather than the description of functions could be emphasized in such a manuscript considering the scope of the conference. This would help the readers understand why it is important to have BrainPy given other simulators.
> >
> > While I am wiling to increase my score, I really hope the authors can friendly consider my suggestions to improve their work.

---

> > > ### Author Response · Authors · 2023-11-22
> > >
> > > Dear kvGz,
> > >
> > > Thank you for reiterating your perspective. You said that ``to contribute broadly to the field beyond a single paper``, this is exactly what we are pursuing. We are fully committed to not only improving the programming framework and engineering quality of BrainPy but also enhancing its modeling algorithms, such as incorporating biologically plausible training methods and memory-efficient parallelization, as well as developing standard large-scale models.
> > >
> > > Your encouragement has given us the further confidence to pursue these ambitious goals. We welcome any additional suggestions or recommendations you may have.
> > >
> > > Best,
> > >
> > > The BrainPy team

---

### Author Response · Authors · 2023-11-22

We greatly appreciate the reviewers for their careful evaluation and constructive feedback throughout the review process. Your insights have helped improve and strengthen our work. In response, we have re-uploaded an updated manuscript and supplementary material incorporating substantial new results in the Appendix, marked in red font. These additions address the reviewers’ comments and lend further support to the claims in our paper. We are grateful for the chance to respond to these constructive suggestions. Please do not hesitate to contact us should you need any clarification or have additional questions during the re-evaluation of our resubmission.

---

### Meta-Review · Area_Chair_yV7J · 2023-12-03

**Metareview:**

The paper presents a software framework for building differentiable brain simulations, based on JAX and XLA. This is an important contribution, as it will make it possible to task-train biophysically realistic simulations of neural networks, thereby closing the gap between 'deep learning' and 'computational neuroscience'. It is a bit of an unusual ICLR paper (in that it mainly provides a software package, not new theories or empirical analyses), but the consensus of the reviewers was nevertheless that this is a highly useful contribution.

**Justification For Why Not Higher Score:**

I am mostly going with 'poster' because this is a software package, not a 'scientific result'. I really am not quite sure on that, and one might want to bump it up.

**Justification For Why Not Lower Score:**

Clear consensus to accept, and I see now reason to overrule it.

---

### Decision · Program_Chairs · 2024-01-16

Accept (poster)